# Extracellular vesicle-mediated release of bis(monoacylglycerol) phosphate is regulated by LRRK2 and glucocerebrosidase activity

Elsa Meneses-Salas[1†], Moises Castellá[1†], Marianna Arnold[2], Frank Hsieh[2], Rubén Fernández-Santiago[3,4], Mario Ezquerra[3,4], Alicia Garrido[3,4], María-José Martí[3,4], Carlos Enrich[1], Suzanne R Pfeffer[5], Kalpana Merchant[6*], Albert Lu[1*]

[1]Departament de Biomedicina, Unitat de Biologia Cel·lular, Facultat de Medicina i Ciències de la Salut, Centre de Recerca Biomèdica CELLEX, Institut d'Investigacions Biomèdiques August Pi i Sunyer, Universitat de Barcelona, Barcelona, Spain; [2]NextCea Inc, Woburn, United States; [3]Lab of Parkinson Disease and Other Neurodegenerative Movement Disorders, Institut d'Investigacions Biomèdiques August Pi i Sunyer (IDIBAPS), Institut de Neurociències, Universitat de Barcelona, Barcelona, Spain; [4]Parkinson Disease and Movement Disorders Unit, Neurology Service, Institut Clínic de Neurociències, Hospital Clínic de Barcelona, Barcelona, Spain; [5]Department of Biochemistry, Stanford University, Stanford, United States; [6]Department of Neurology, Northwestern University, Chicago, United States

*For correspondence:
kalpana.merchant@
northwestern.edu (KM);
albertlu@ub.edu (AL)

†These authors contributed
equally to this work

Competing interest: See page
20

Reviewing Editor: Ishier Raote,
Institut Jacques Monod, France

## eLife Assessment

This **useful** study presents the potentially interesting idea that LRRK2 regulates cellular BMP levels and their release via extracellular vesicles, with GCase activity further modulating this process in mutant LRRK2-expressing cells. However, some of the evidence supporting these conclusions remains **incomplete**, and additional work is suggested under certain conditions. Overall, the study will be of interest to cell biologists working on Parkinson's disease.

**Abstract** The endolysosomal phospholipid bis(monoacylglycerol)phosphate (BMP) is aberrantly elevated in urine from Parkinson's patients carrying mutations in leucine-rich repeat kinase 2 (LRRK2) and glucocerebrosidase (GCase). Because BMP resides on, and regulates biogenesis of, endolysosomal intralumenal membranes that become extracellular vesicles (EVs) upon release, we hypothesized that increased urinary BMP reflects enhanced exocytosis of BMP-enriched EVs. We analyzed BMP metabolism and EV-associated BMP release in wild-type (WT) and R1441G LRRK2 mouse embryonic fibroblasts (MEFs). Immunofluorescence and transmission electron microscopy revealed structural alterations in endolysosomes and the antibody-accessible BMP pool, indicating disrupted endolysosomal homeostasis. Biochemical analysis of isolated EV fractions showed increased release of LAMP2-positive EVs by mutant cells, partially restored by LRRK2 kinase inhibition but further, variably, increased by GCase inhibition. Mass spectrometry detected higher total di-22:6-BMP and di-18:1-BMP in mutant LRRK2 MEFs compared to WT. Inhibition of LRRK2 partially restored cellular BMP, whereas GCase inhibition further elevated it. In EVs from mutant cells, LRRK2 inhibition reduced BMP content, while GCase inhibition tended to increase it. Metabolic labeling showed elevated BMP was not due to increased synthesis, despite higher levels of the BMP-synthesizing

enzyme CLN5 in mutant MEFs and patient fibroblasts. Finally, pharmacological modulation of EV release and live total internal reflection fluorescence imaging in human G2019S LRRK2 fibroblasts further confirmed that BMP release is likely associated with EV secretion. Together, these results establish LRRK2 as a regulator of BMP in cells and its release through EVs and suggest that GCase activity further modulates this process in LRRK2 mutant cells. Mechanistic insights from these studies have implications for the use of BMP-positive EVs as potential biomarkers for Parkinson's disease.

## Introduction

Variants in the gene encoding leucine-rich repeat kinase 2 (*LRRK2*) and β-glucocerebrosidase (GCase; *GBA1*) together constitute the largest contributors to familial and sporadic Parkinson's disease (PD) (*Smith et al., 2022*). Although G2019S is the most prevalent PD-associated *LRRK2* mutation, R1441G is more penetrant and is associated with an earlier disease onset (*Yue and Lachenmayer, 2011*; *Pont-Sunyer et al., 2017*; *Fan et al., 2021*). A shared mechanism of pathogenesis among all PD-causing LRRK2 mutations is increased LRRK2 kinase activity, which leads to phosphorylation of its cognate Rab GTPase substrates, including Rab8, Rab10, and Rab12, that are key regulators of intracellular vesicle trafficking (*Pfeffer, 2023*). A prevailing hypothesis of pathogenesis in LRRK2 PD cases is that LRRK2 phosphorylation of Rabs results in defective autophagy and endolysosomal homeostasis and defective ciliogenesis in the nigrostriatal circuit (*Pfeffer, 2023*; *Roosen and Cookson, 2016*).

*GBA1* encodes a lysosomal hydrolase, glucocerebrosidase (GCase), essential for glycosphingolipid degradation. Homozygous loss-of-function mutations in *GBA1* cause Gaucher's disease, a lysosomal storage disorder characterized by cytotoxic accumulation of lipid substrates of GCase, glucosylceramide, and glucosylsphingosine; heterozygous carriers have a 5- to 10-fold increased risk of PD (*Sidransky and Lopez, 2012*; *Duran et al., 2013*). The predominant hypothesis for the pathogenic mechanism of *GBA1* variants is related to aberrant aggregation and clearance of alpha-synuclein in a feed-forward cascade (*Fredriksen et al., 2021*; *Stojkovska et al., 2022*); a hypothesis being tested clinically using GCase activators. Additionally, we and others previously showed that GCase enzymatic function is modulated by LRRK2 kinase activity (*Ysselstein et al., 2019*; *Kedariti et al., 2022*). Altogether, it is possible that dysregulation of the endolysosomal biology is associated with pathogenic variants of both *LRRK2* and *GBA1* (*Roosen and Cookson, 2016*; *Do et al., 2019*).

Bis(monoacylglycerol)phosphate (BMP, also known as LBPA or lysobisphosphatidic acid) is an atypical, negatively charged glycerophospholipid important for maintaining endolysosomal homeostasis (*Gruenberg, 2020*). BMP resides primarily in endolysosomal intralumenal vesicles (ILVs) where it participates in multiple processes, including cholesterol egress, exosome biogenesis, and lipid catabolism (*Gruenberg, 2020*; *Gallala and Sandhoff, 2011*). Of note, BMP acts together with Saposin-C as enzymatic cofactors of GCase (*Alattia et al., 2007*; *Abdul-Hammed et al., 2017*). Multiple BMP isomers are present in cells, based on differences in their fatty acyl chains and their positions relative to the sn-2 and sn-3 carbon atoms on each of its two glycerol backbones (*Gruenberg, 2020*; *Gallala and Sandhoff, 2011*); the sn-2:sn-2′ isomer is the proposed active form (*Kobayashi et al., 1998*; *Matsuo et al., 2004*). Dioleoyl-BMP (di-18:1-BMP) is the predominant form in numerous cell lines analyzed (*Mason et al., 1972*; *Huterer and Wherrett, 1979*; *Bouvier et al., 2009*; *Grabner et al., 2020*) and didocosahexaenoyl-BMP (di-22:6-BMP) is one of the most abundant species in the brain (*Akgoc et al., 2015*; *Boland et al., 2022*). Fatty acyl composition is thought to influence BMP biophysical properties and function (*Goursot et al., 2010*).

We and others recently reported elevated levels of total di-18:1-BMP and total di-22:6-BMP species in the urine of carriers of PD-associated *LRRK2* and *GBA1* mutations (*Alcalay et al., 2020*; *Merchant et al., 2023*; *Gomes et al., 2023*). Although we did not observe a correlation between urinary BMP and PD progression (*Merchant et al., 2023*), these studies underscore BMP's utility as a patient enrichment and target modulation biomarker in therapeutic trials (*Jennings et al., 2023*). Based on our previous findings, in the present study, we focused on these two BMP species.

Given the specific intracellular localization of BMP (in ILVs; exosomes when released) and its proposed roles in exosome biogenesis (*Matsuo et al., 2004*), we hypothesized that its increased levels in PD biofluids may be a consequence of an increase in the secretion of BMP containing extracellular vesicle (EV). To gain mechanistic insight into the regulation and biological significance of BMP release, we monitored BMP release in EVs and performed kinetic analyses of BMP biosynthesis in

mouse embryonic fibroblasts (MEFs) from wild-type (WT) or R1441G LRRK2 knock-in mice. Moreover, endolysosomal exocytosis was visualized and quantified in skin fibroblasts derived from both healthy donors and G2019S LRRK2 PD patients. In addition, we examined the regulation of BMP by a LRRK2 kinase inhibitor, MLi-2, and a GCase inhibitor, conduritol β-epoxide (CBE), to gain insights into the role of BMP in potential therapeutic vs. pathophysiological responses, respectively. Our data indicate that LRRK2 kinase activity modulates BMP release in EVs and further suggests that GCase function may contribute to this process.

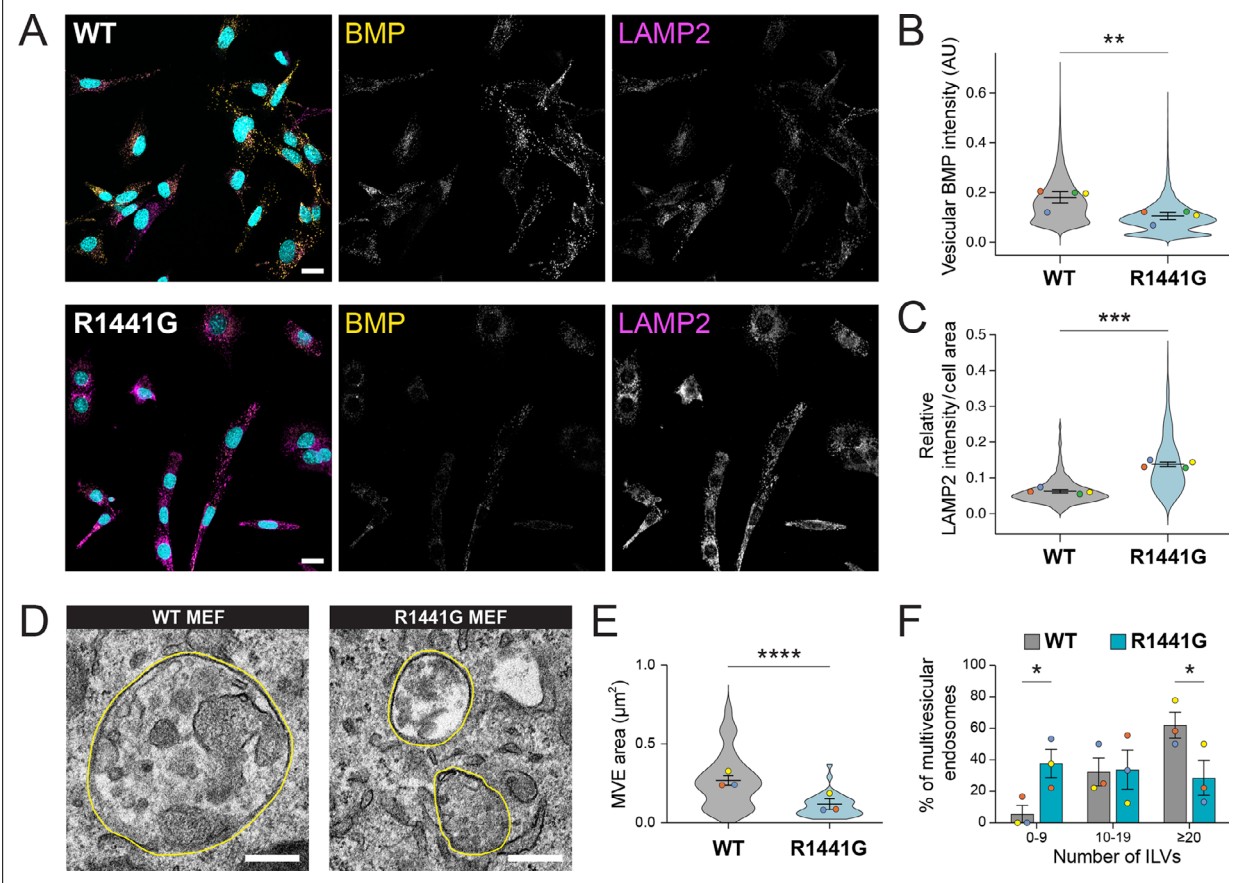

**Figure 1.** Alterations in antibody-accessible BMP and endolysosomal morphology in R1441G LRRK2 mouse embryonic fibroblast (MEF) cells. (**A**) Confocal microscopy of endogenous BMP (yellow) and LAMP2 (magenta) immunofluorescence in wild-type (WT) and R1441G LRRK2 MEFs. Scale bar: 20 μm. Quantification of vesicular BMP intensity (**B**) and LAMP2 relative intensity (**C**) per cell area. Colored dots represent mean value from four independent experiments, and violin plots show the distribution of individual cell data. Significance determined by two-tailed paired *t*-test **p < 0.01, ***p < 0.001. (**D**) Representative transmission electron microscopy (TEM) images of multivesicular endosomes (MVE) from WT and R1441G LRRK2 MEFs. MVB periphery highlighted in yellow. Scale bar: 250 nm. (**E**) MVE area (μm²) quantification in WT and R1441G LRRK2 mutant cells. Colored dots represent mean values from 3 independent experiments and violin plots show the distribution of individual cell data (35–45 cells/group). (**F**) Quantification of intraluminal vesicles (ILVs) per MVE in WT and R1441G LRRK2 MEF cells. The number of ILVs per MVE is binned in three groups and plotted as a percentage of MVE from the total population of each experiment independently. Data from 3 independent experiments (mean ± SEM). Significance determined by two-tailed unpaired *t*-test (**E**) and ordinary two-way ANOVA, uncorrected Fisher's LSD (**F**). *p < 0.05, ****p < 0.0001.

The online version of this article includes the following source data for figure 1:

**Source data 1.** IF images in panel A.

**Source data 2.** TEM images in panel D.

**Source data 3.** Plotted values in panels B, C, E, F.

## Results

### Analysis of BMP-positive endolysosomes in R1441G LRRK2 MEFs

We first performed immunofluorescence microscopy of BMP-positive endolysosomes in both WT and R1441G LRRK2 knock-in MEFs. The overall antibody-accessible BMP pool was significantly decreased in mutant LRRK2-expressing cells compared to WT (*Figure 1A, B*). In contrast, endolysosomal LAMP2 fluorescence was higher in mutant LRRK2 MEFs (*Figure 1A, C*). Previous studies found structural defects in endolysosomes of different cell types harboring pathogenic LRRK2 mutations (*Henry et al., 2015*; *Hockey et al., 2015*; *de Rus Jacquet et al., 2021*). Since BMP is specifically enriched in ILVs in the endolysosomes where it regulates important catabolic reactions, we examined multivesicular endosome (MVE) morphology of these cells. In agreement with previous findings (*de Rus Jacquet et al., 2021*), transmission electron microscopy revealed decreased MVE area in R1441G LRRK2 MEFs (average area = 0.11 $\mu m^2$) compared to WT MEFs (average area = 0.27 $\mu m^2$) (*Figure 1D, E*). In addition, the overall endolysosomal ILV number was diminished in the R1441G LRRK2 cell line (*Figure 1F*). These data suggest that the pathogenic LRRK2 mutation alters BMP-positive endolysosome morphology and ILV content.

### Investigating the impact of LRRK2 and GCase activities on EV release modulation

Given the previously described roles of BMP in ILV/exosome biogenesis, we investigated whether the alterations observed in antibody-accessible BMP and MVE ILV number in LRRK2 mutant MEFs could be explained by changes in EV release. We isolated and characterized EVs from both WT and R1441G MEFs and assessed the effects of LRRK2 and GCase pharmacological inhibitors, MLi-2 and CBE, respectively, on EV content and number. Inhibition of LRRK2 kinase and GCase enzymatic activities was confirmed, respectively, by monitoring phospho-Rab10 levels in whole cell lysates (WCL) and performing a fluorescence-based GCase activity assay in cells (*Figure 2A, B*). Consistent with our immunofluorescence data, upregulation of LAMP2 was observed in R1441G LRRK2 WCL (*Figure 2A, C, D, G, H*), as previously reported in LRRK2 G2019S knock-in mouse brain (*Albanese et al., 2021*). MLi-2 treatment for 48 hr partially reversed this phenotype (*Figure 2A, C, D*), suggesting that aberrant LRRK2 kinase activity influences endolysosomal homeostasis.

We next analyzed the profiles of isolated EV fractions from both WT and R1441G LRRK2 mutant MEFs. Both LAMP2 and Flotillin-1 were assessed to explore alterations in release of EV subpopulations; LAMP2 is enriched in ILV-derived EVs while Flotillin-1 is also seen in plasma membrane-derived ectosomes that reflect outward budding of the plasma membrane (*Kowal et al., 2016*; *Mathieu et al., 2021*; *Ferreira et al., 2022*). Biochemical analysis revealed an elevation of LAMP2, but not Flotillin-1, in EVs derived from R1441G LRRK2 MEFs compared to those from WT MEFs (*Figure 2C, E, F, G, I, J*). Upon MLi-2 treatment, a modest but significant decrease in EV-associated LAMP2 was observed in isolated EV fractions from mutant LRRK2 MEFs (*Figure 2C, E*). In contrast, inhibition of GCase activity in mutant LRRK2 cells led to an increase in LAMP2, but not Flotillin-1, in isolated EVs compared to those derived from WT MEFs (*Figure 2G, I, J*). Finally, analysis of WT cell-derived EVs revealed no major differences in EV marker levels between untreated (control) and MLi-2- or CBE-treated conditions (*Figure 2—figure supplement 1A–C*).

To complement these data, we conducted nanoparticle tracking analysis (NTA). Isolated EV fractions from WT and R1441G LRRK2 cells exhibited comparable particle size distributions (*Figure 2—figure supplement 1D*). Treatments with MLi-2 and CBE yielded measurable quantitative changes in EV concentrations that, despite not reaching statistical significance, gave similar trends as those seen in our biochemical analyses (*Figure 2—figure supplement 1E*), potentially reflecting the inherent variability of NTA due to its inability to distinguish EVs from non-vesicular particles (*Gardiner et al., 2013*; *Bachurski et al., 2019*). For the R1441G MEF cells, MLi-2 decreased EV concentration while CBE increased EV particles per ml, in agreement with the effects observed biochemically.

Altogether, these results suggest that EV secretion is influenced by LRRK2 kinase and, to a more variable degree, by GCase hydrolase activities; while MLi-2 treatment decreases EV release, CBE appears to increase it.

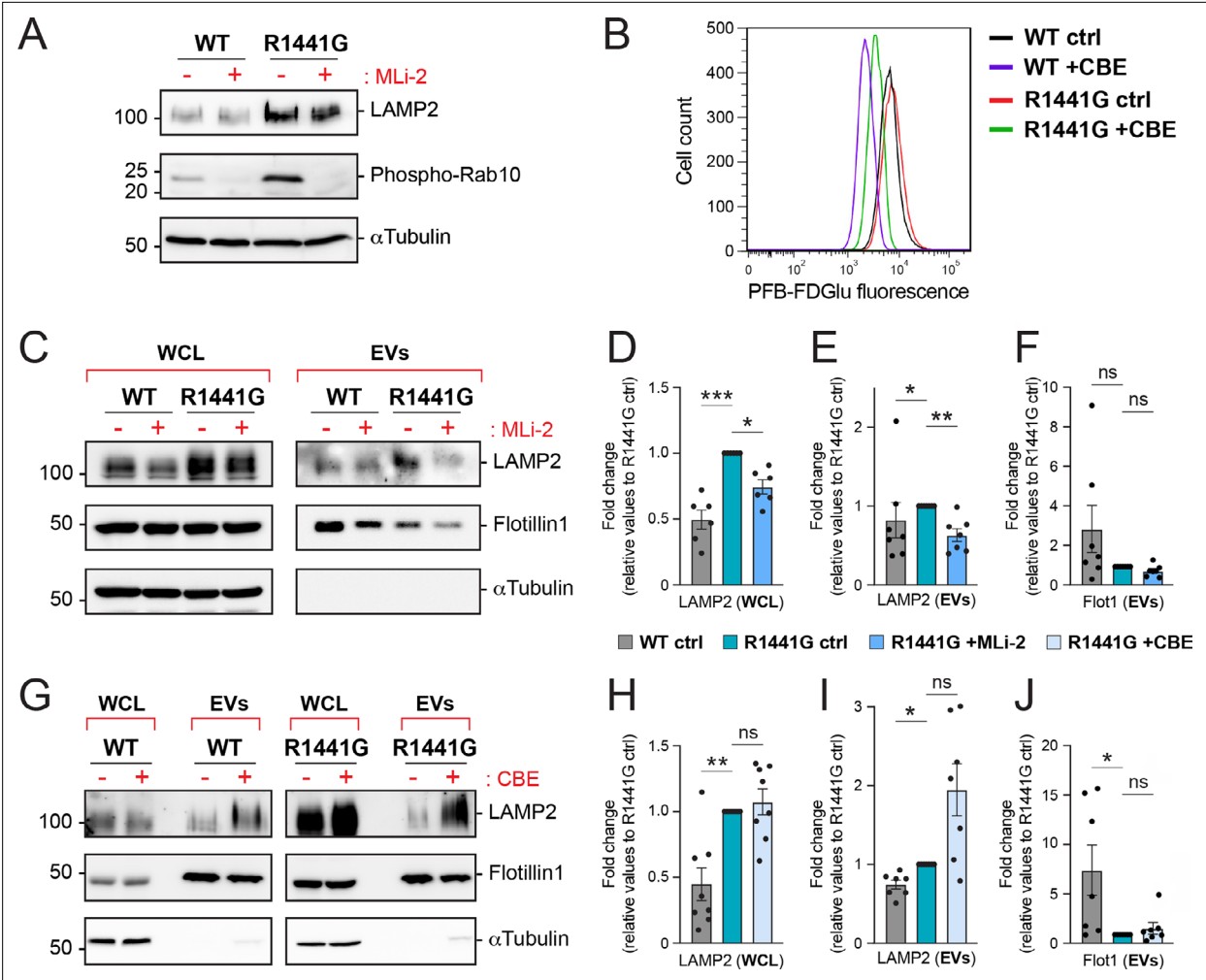

**Figure 2.** LRRK2 and GCase activities modulate extracellular vesicle (EV) production. (**A**) Whole cell lysates (WCL) from wild-type (WT) and R1441G LRRK2 mouse embryonic fibroblast (MEF) cells treated with 200 nM MLi-2 for 24 hr were analyzed by immunoblotting. Representative images of LAMP2, phospho-Rab10, and α-Tubulin levels are shown. Molecular weight marker mobility is shown in kDa. (**B**) Flow cytometry measurement of GCase activity using PFB-FDGlu fluorescent GCase substrate in WT and R1441G LRRK2 mutant MEF cells treated with 300 μM conduritol β-epoxide (CBE) for 24 hr. WCL and isolated EVs from WT and R1441G LRRK2 mutant MEF cells treated with 200 nM MLi-2 (**C**) or 300 μM CBE (**G**) for 48 hr were analyzed by immunoblotting. Representative images of LAMP2, Flotillin-1, and α-Tubulin levels are shown. Molecular weight marker mobility is shown in kDa. Immunoblots for LAMP2 and Flotillin-1 in EV fractions required longer exposure times to visualize clear signals across all conditions. Quantification of LAMP2 and Flotillin-1 levels relative to R1441G LRRK2 MEF cells in WCL (**D**, **H**) and isolated EVs (**E**, **F**, **I**, **J**). Data from 6 to 8 independent experiments (mean ± SEM). Significance determined by Kruskal–Wallis test followed by an uncorrected Dunn's post hoc test compared to R1441G LRRK2 control *p < 0.05, **p < 0.01, ***p < 0.001; ns, not significant.

The online version of this article includes the following source data and figure supplement(s) for figure 2:

**Source data 1.** Uncropped blots.

**Source data 2.** Annotated uncropped blots.

**Source data 3.** Plotted values in panels D–F, H–J.

**Figure supplement 1.** Further characterization of mouse embryonic fibroblast (MEF)-derived extracellular vesicle (EV) fractions.

**Figure supplement 1—source data 1.** Plotted values in panels A–C, E.

## Targeted BMP lipid analysis in intact cells and EVs

Ultra-performance liquid chromatography–tandem mass spectrometry (UPLC–MS/MS) was used to measure quantitatively the impact of LRRK2 and GCase activities on BMP isoform abundance in cells and isolated EV fractions. Remarkably, an overall increase in total BMP isoforms was detected in mutant LRRK2 MEF cell lysates (*Figure 3A–D*), with di-22:6-BMP and di-18:1-BMP as the major

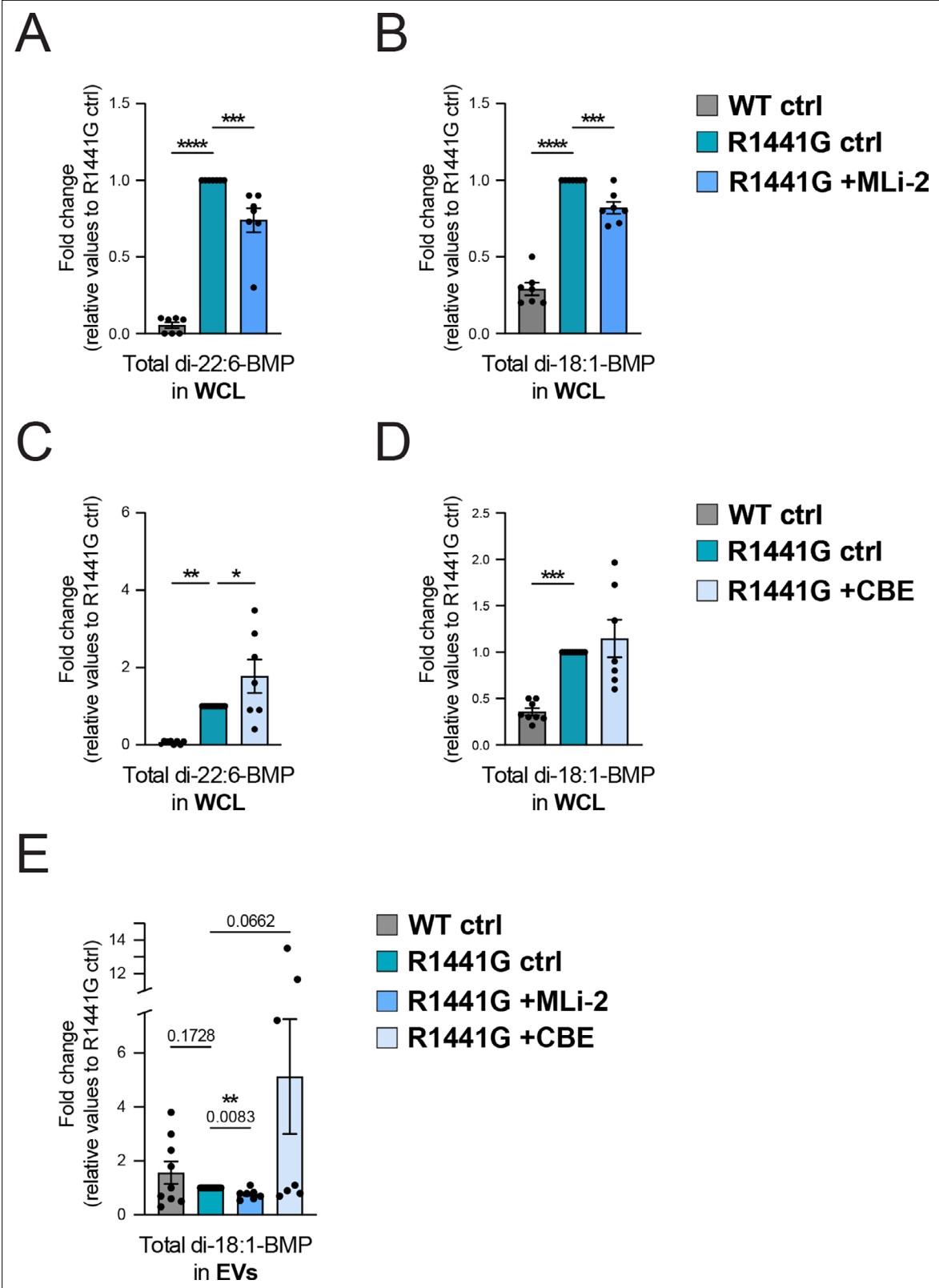

**Figure 3.** Targeted lipid pathway analysis of BMP abundance in cellular and isolated extracellular vesicle (EV) fractions. Ultra-performance liquid chromatography–tandem mass spectrometry (UPLC–MS/MS) determination of BMP isoforms normalized to protein content from cells treated with 200 nM MLi-2 (**A, B**) or 300 μM conduritol β-epoxide (CBE) (**C, D**) for 48 hr. Data shown as fold change relative to untreated R1441G LRRK2 mouse embryonic fibroblast (MEF) cells. Only BMP isoforms that were detected are shown. (**E**) UPLC–MS/MS determination of BMP isoforms normalized to

*Figure 3 continued on next page*

*Figure 3 continued*

protein content in EVs isolated from cells treated with 200 nM MLi-2 or 300 μM CBE for 48 hr. Only BMP isoforms that were detected are shown. Data from 3 to 6 independent experiments (mean ± SEM). Significance determined by ordinary one-way ANOVA, uncorrected Fisher's LSD (**A–D**) and one-way ANOVA with the Geisser–Greenhouse correction, uncorrected Fisher's LSD (**E**). *p < 0.05, **p < 0.01, ***p < 0.001, ****p < 0.0001.

The online version of this article includes the following source data and figure supplement(s) for figure 3:

**Source data 1.** Plotted values in panels A–E.

**Figure supplement 1.** BMP levels are unchanged in wild-type (WT) mouse embryonic fibroblasts (MEFs) following MLi-2 or conduritol β-epoxide (CBE) treatment.

**Figure supplement 1—source data 1.** Plotted values in panels A, B, D.

**Figure supplement 2.** Quantitative analysis of GCase substrates in mouse embryonic fibroblast (MEF) cells and extracellular vesicle (EV) fractions.

**Figure supplement 2—source data 1.** Plotted values in panels A, B.

species. It is important to note that mass spectrometry-based methods detect total BMP content while antibody staining (*Figure 1A, B*) only detects so-called 'antibody-accessible' BMP (*Lu et al., 2022*). The mass spectrometry-based increase in total BMP in LRRK2 mutant cells indicates that this pool is less antibody-accessible than that present in WT cells. Alternately, the anti-BMP antibody may be less specific and detect other analytes.

In R1441G cells, LRRK2 inhibition for 48 hr with MLi-2 decreased total di-22:6-BMP and total di-18:1-BMP levels by ~20% (*Figure 3A, B*). Conversely, inhibition of GCase increased total BMP levels, reaching significance for di-22:6-BMP levels compared with untreated, R1441G LRRK2 cells (*Figure 3C*). No statistically significant differences in intracellular BMP levels were observed in WT LRRK2 MEFs upon LRRK2 or GCase inhibition (*Figure 3—figure supplement 1A, B*), suggesting a dominant role of mutant LRRK2 activity in BMP regulation. Overall, di-oleoyl (18:1)- and di-docosa-hexaenoyl (22:6)-BMP species were the most abundant in MEFs, whereas other isoforms, such as di-arachidonoyl (20:4)- and di-linoleoyl (18:2)-BMP, were present at lower levels but were also consistently elevated in mutant LRRK2 MEFs (*Figure 3—figure supplement 1C*). These data suggest that there are LRRK2-independent clonal differences leading to differences in basal BMP content between WT and mutant MEF cells; however, such clonal variation does not impact the effect of MLi-2 or CBE treatment in R1441G cells.

Analysis of isolated EV fractions detected only the di-18:1-BMP isoform, with no evidence of di-22:6-BMP. Although a trend toward higher di-18:1-BMP levels was observed in EVs derived from WT cells compared to those from mutant LRRK2 cells, this difference was not statistically significant (*Figure 3E*). Treatment of LRRK2 R1441G MEFs with MLi-2 resulted in a significant partial decrease in EV-associated total BMP (*Figure 3E*), consistent with previous observations in non-human primates and PD mouse models that showed decreased extracellular urinary BMP upon LRRK2 kinase pharmacological inhibition (*Fuji et al., 2015*; *Baptista et al., 2020*; *Jennings et al., 2022*). In contrast, inhibition of GCase activity yielded an opposite, albeit not significant, trend (*Figure 3E*). These latter findings may be explained by the observation that GCase inhibition by CBE was less pronounced in LRRK2 R1441G cells compared to WT cells under identical concentration and treatment duration conditions (*Figure 2B*). Finally, no significant differences in EV-associated BMP abundance between untreated (control) and MLi-2- or CBE-treated WT LRRK2 MEFs were observed (*Figure 3—figure supplement 1D*).

In addition to analyzing BMP, we also examined GCase lipid substrates in both cells and isolated EVs. Targeted quantification of sphingolipid species revealed elevated GCase substrate abundance in both CBE-treated WT and mutant LRRK2 cells, validating the treatment paradigm used in this study. Interestingly, CBE-mediated accumulation of GCase substrates was more evident in R1441G LRRK2 than in WT cells (*Figure 3—figure supplement 2A*). This suggests that, despite a less pronounced direct effect on GCase activity (*Figure 2B*), the R1441G mutation might contribute to broader endo-lysosome dysfunction, which could alter the dynamics of substrate accumulation. On the other hand, analysis of isolated EV fractions revealed lower levels of glucosylceramide, galactosylceramide, and glucosylsphingosine in EVs from R1441G LRRK2 cells compared to those from WT MEFs, but no significant differences were detected between control or CBE-treated conditions independent of LRRK2 mutation status (except for glucosylsphingosine, which showed an increase in WT-EVs upon GCase inhibition; *Figure 3—figure supplement 2B*).

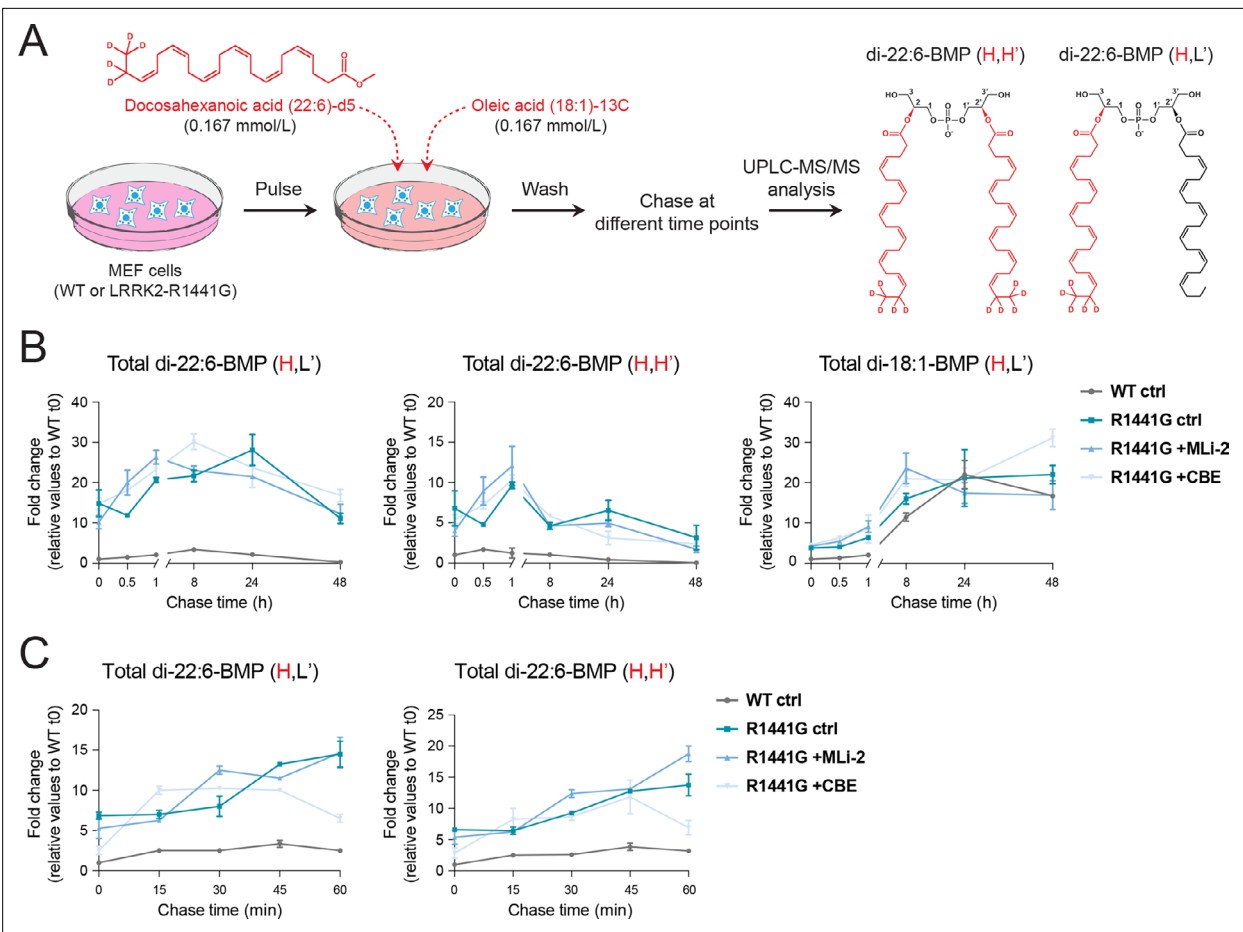

**Figure 4.** Inhibition of LRRK2 or GCase activities does not significantly impact BMP biosynthetic and catabolic rates. (**A**) Schematic representation of the BMP metabolic labeling protocol with deuterated docosahexaenoic acid and [13]C-labeled oleic acid. Wild-type (WT) and R1441G mutant mouse embryonic fibroblast (MEF) cells were incubated with a pulse of DHA-d5/OA-[13]C for 20 min, followed by a chase for different times. Cells were then collected for subsequent BMP lipidomic analysis. Structures of unlabeled (L) and isotope-labeled (H) fatty acids are shown in black or red, respectively. (**B, C**) Ultra-performance liquid chromatography–tandem mass spectrometry (UPLC–MS/MS) determination of BMP isoforms normalized to protein content from WT MEF cells and R1441G LRRK2 mutant MEF cells ± MLi-2 (200 nM) or conduritol β-epoxide (CBE) (300 μM). Long (**B**) and short (**C**) chase time points shown as fold change relative to WT control MEF cells time 0. Only BMP isoforms that were detected are shown. Data from 3 replicated experiments (mean ± SEM).

The online version of this article includes the following source data for figure 4:

**Source data 1.** Plotted values in panels B, C.

Altogether, given that BMP is specifically enriched in ILVs (which become exosomes upon release), the data presented above support our biochemical analysis (*Figure 2C, E, G, I*) and suggest that LRRK2 activity regulates BMP release in association with LAMP2-positive exosomes, whereas GCase activity appears to have a more variable effect under the tested conditions.

## BMP biosynthesis is not influenced by LRRK2 kinase and GCase activities

To rule out the possibility that differences in cellular and EV-associated BMP levels following MLi-2 or CBE treatment of R1441G cells were caused by alterations in BMP metabolism rather than changes in membrane trafficking and EV release, we performed metabolic labeling experiments using heavy (H) isotope-labeled 22:6 and 18:1 fatty acids as BMP precursors.

Cells were pulsed for 25 min with heavy isotope BMP precursors and then washed and chased for different time points (*Figure 4A*). UPLC–MS/MS analysis allowed us to differentiate unlabeled versus semi-labeled (H,L'; only one heavy isotope-labeled fatty acid chain) or fully labeled (H,H'; both fatty

acid chains heavy isotope-labeled) BMP species (*Figure 4A*). Initial experiments were performed over 48 hr using mutant LRRK2 cells under control and LRRK2- or GCase-inhibition conditions; untreated WT cells were included for comparison. Strikingly, even at time 0 and throughout the length of the experiment, R1441G LRRK2 cells displayed substantially higher H,L'- and H,H'-22:6-BMP species than WT cells (left and middle plots in *Figure 4B*). H,L', but not H,H' 18:1-BMP was detected, which appeared to increase to a greater extent in R1441G LRRK2 cells at least during the initial 8 hr of the experiment (right plot, *Figure 4B*). Despite these differences between WT and mutant LRRK2 cells, neither inhibition of LRRK2 or GCase had a major impact on the kinetics of BMP synthesis or catabolism (*Figure 4B*).

In these experiments, semi-labeled BMP species gradually increased for up to 24 hr. This likely reflects re-utilization of isotope-labeled lysophosphatidylglycerol and fatty acid precursors generated after degradation of either H,L' or H,H' BMP by endolysosomal hydrolase PLA2G15 (*Nyame et al., 2025*). To better resolve initial BMP synthesis rates, we reduced the pulse labeling time and chase time to 60 min. In this set of experiments, only isotope-labeled 22:6-BMP species were detected (*Figure 4C*), possibly reflecting a preference for 22:6 BMP synthesis by MEFs. As before, even at time 0 and throughout the 60 min duration of the assay, 22:6-BMP levels were consistently higher in R1441G LRRK2 cells compared with WT MEFs. Again, no overall rate differences were seen between untreated and MLi-2- or CBE-treated cells. These experiments support the hypothesis that rather than altering BMP metabolic rates, LRRK2 and GCase activities influence EV-mediated BMP release.

## Increased BMP synthase protein expression in mutant LRRK2 cells

To further investigate the biological significance of BMP upregulation observed in mutant LRRK2 MEF cells (*Figures 3A-D and 4B, C*), we examined the expression of CLN5, a key lysosomal enzyme in the BMP biosynthetic pathway (*Medoh et al., 2023*; *Bulfon et al., 2024*). Interestingly, biochemical analysis of total cell lysates revealed a significant fold-change increase in CLN5 protein levels in R1441G LRRK2 MEFs relative to WT cells (*Figure 5A*). Notably, a 16-hr treatment with MLi-2 reduced CLN5 levels to a comparable fold-change extent (*Figure 5B*). To validate these findings in a human and disease-relevant context, we examined CLN5 protein expression in cultured fibroblasts derived from PD patients carrying the G2019S LRRK2 mutation. Consistent with our observations in MEF cells, CLN5 protein levels were reduced following 16 hr MLi-2 treatment in a concentration-dependent manner (*Figure 5C*). Notably, the maximal reduction was observed at the same 200 nM MLi-2 concentration used in MEF cell experiments, a dose that also did not induce observable cytotoxic effects in human fibroblasts. Similar results were obtained in patient-derived fibroblasts harboring the R1441G LRRK2 mutation (*Figure 5D*). Altogether, these data suggest that LRRK2 kinase activity may regulate CLN5 protein expression. The upregulation of CLN5 may be due to an overall upregulation of lysosomal enzymes as LAMP2 levels were also increased (*Figure 2A, C, D, G, H*). Moreover, the observed baseline differences in BMP, as documented in the flux study above (*Figure 4B, C*), could result from CLN5 upregulation. The lack of significant changes in the BMP synthesis rate (*Figure 4B, C*) suggests either a limitation in substrate availability or that CLN5 is operating at maximal capacity.

## BMP release is EV-mediated

Our data suggest that BMP is exocytosed in association with EVs and that LRRK2 and GCase activities modulate BMP secretion. To determine the magnitude of BMP release via EV secretion, we assessed the impact of pharmacological modulators of EV release. Treatment of WT MEFs with GW4869, a selective type 2-neutral sphingomyelinase inhibitor, decreased EV release as monitored by reduced levels of LAMP2 and Flotillin-1 in EV fractions (*Figure 6A*). Under these conditions, we observed a parallel decrease in exosomal BMP (*Figure 6A, B*). In contrast, enhancing EV release with bafilomycin A1 (B-A1), a pharmacological inhibitor of the endolysosomal proton pump V-ATPase that dramatically boosts EV exocytosis (*Mathieu et al., 2021*; *Lu et al., 2018*), resulted in the opposite trend (*Figure 6A, C*): EV markers and BMP levels increased. Biochemical analysis of EV release in R1441G LRRK2 cells showed similar behavior as in WT MEFs upon treatment with these agents (*Figure 6A*, *Figure 6—figure supplement 1*). GW4869 inhibition of EV release increased cellular BMP content (*Figure 6D*), while enhanced EV release upon B-A1 treatment diminished intracellular total di-22:6-BMP and di-18:1-BMP (*Figure 6E*). Altogether, these results strongly support the notion that BMP is released by EV-mediated MVE exocytosis.

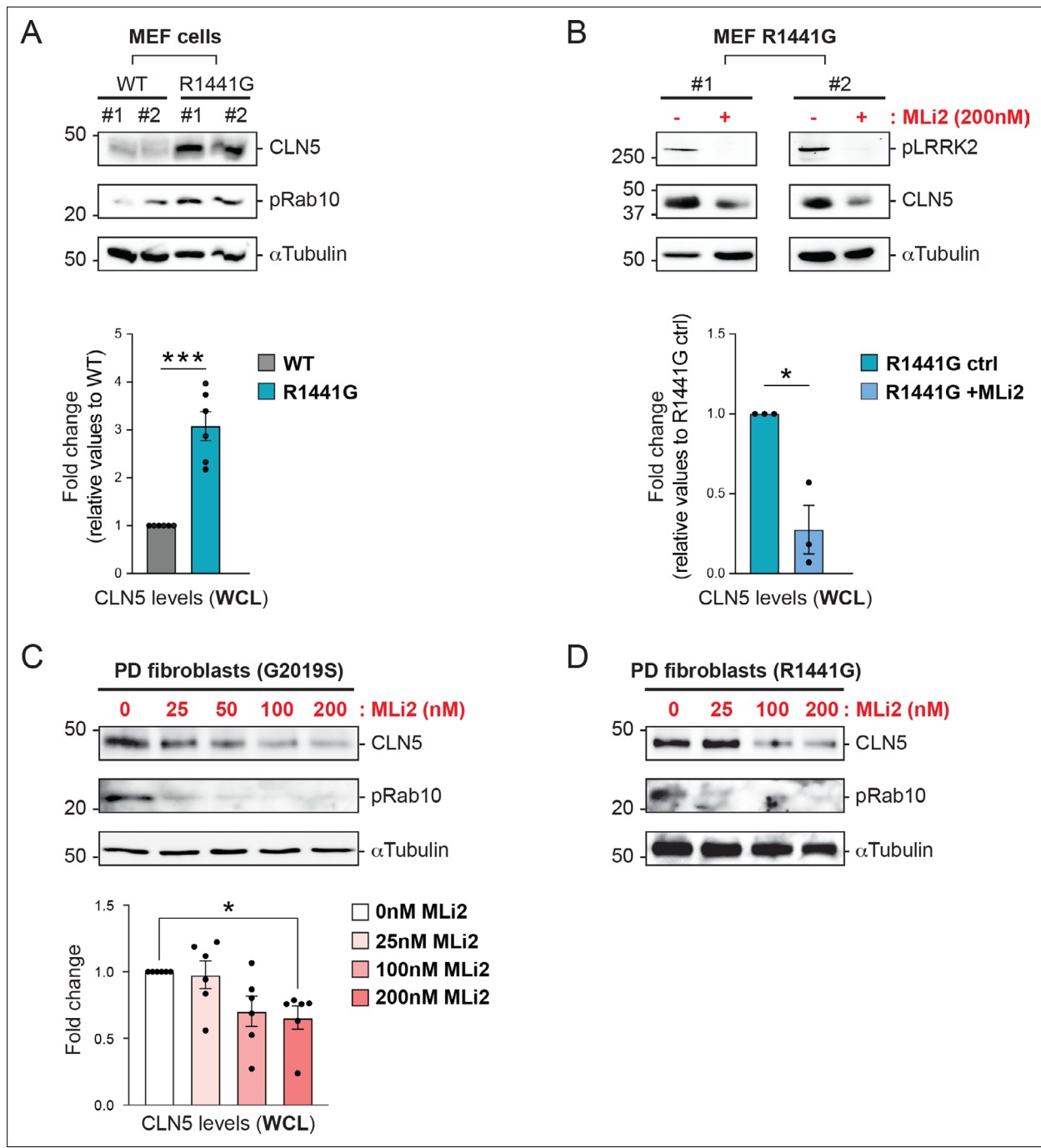

**Figure 5.** LRRK2 activity modulates CLN5 expression levels. (**A**) Whole cell lysates (WCL) from wild-type (WT) and R1441G LRRK2 mouse embryonic fibroblast (MEF) cells were analyzed by immunoblotting. Representative immunoblots of CLN5, phospho-Rab10 (pRab10), and α-Tubulin are shown from two (#1 and #2) out of six independent experiments. Molecular weight marker mobility is shown in kDa. Plot at the bottom shows quantification of CLN5 immunoblot levels (relative to WT). (**B**) WCL from R1441G LRRK2 MEF cells treated with 200 nM MLi-2 for 16 hr were analyzed by immunoblotting. Representative immunoblots of phospho-LRRK2 (pLRRK2), CLN5, and α-Tubulin levels are shown from two (#1 and #2) out of three independent experiments. Molecular weight marker mobility is shown in kDa. Plot at the bottom shows quantification of CLN5 immunoblot levels in WCL of R1441G LRRK2 MEF cells, untreated (ctrl) or treated with MLi-2. (**C**) WCL from G2019S LRRK2 patient-derived fibroblasts treated with indicated increasing MLi-2 concentrations for 16 hr were analyzed by immunoblotting. Immunoblots of CLN5, phospho-Rab10 (pRab10), and α-Tubulin levels are shown from one representative experiment (*n* = 6 per condition, obtained from two independent replicate experiments using fibroblast cell lines derived from three different patients). Molecular weight marker mobility is shown in kDa. Plot at the bottom shows quantification of CLN5 immunoblot levels relative to G2019S LRRK2 patient-derived fibroblasts treated with MLi-2 at the indicated concentrations. (**D**) WCL from R1441G LRRK2 patient-derived fibroblasts treated with indicated increasing MLi-2 concentrations for 24 hr were analyzed by immunoblotting. Immunoblots of CLN5, phospho-Rab10 (pRab10),

*Figure 5 continued on next page*

*Figure 5 continued*

and α-Tubulin levels are shown from one representative experiment. Significance in (**A**) and (**B**) determined by two-tailed, unpaired *t*-test; significance in (**C**) determined by Dunnett's one-way ANOVA test; *p < 0.05, ***p < 0.001.

The online version of this article includes the following source data for figure 5:

**Source data 1.** Uncropped blots.

**Source data 2.** Annotated uncropped blots.

**Source data 3.** Plotted values in panels A–C.

## Monitoring in vivo endolysosomal exocytosis and EV release in patient-derived G2019S LRRK2 fibroblasts

All our previous experiments investigating the role of LRRK2 in EV release regulation were conducted using mouse-derived cells. To further validate these findings, we employed skin fibroblasts derived from healthy donors (control) and G2019S LRRK2 PD patients. We first performed an immunofluorescence and biochemical analysis of the BMP-positive endolysosomal compartment in three independent skin fibroblast cell lines from each cohort. Consistent with our observations in MEFs, we observed a decrease in BMP immunostaining in LAMP1-positive endolysosomes of G2019S LRRK2 cells compared to control (*Figure 7A, B*). LAMP1 levels remained unchanged between the two cohorts (*Figure 7A, C*).

Next, we biochemically characterized endolysosomes isolated by subcellular fractionation on discontinuous sucrose gradients. Interestingly, and consistent with previous studies reporting association of LRRK2 with isolated endolysosomes (*Bentley-DeSousa et al., 2025*), a pool of LRRK2 was detected in fractions positive for CLN5 and the endolysosomal marker CD63, but negative for α-Tubulin (*Figure 7—figure supplement 1A*). As seen in our WCL analysis (*Figure 5C*), relative CLN5 levels appeared increased in G2019S LRRK2 fibroblasts (*Figure 7—figure supplement 1A*, bottom fractions). However, the relative amount of CLN5 co-peaking with isolated CD63-positive endolysosomes was comparable to that in control cells (*Figure 7—figure supplement 1A*, endolysosome fractions). These results could suggest an altered subcellular distribution of CLN5 in R1441G cells. Such mislocalization could contribute to the dissociation between increased CLN5 protein levels (*Figure 5A*) and unchanged BMP biosynthetic rates (*Figure 4B, C*) observed in mutant cells. Nevertheless, it is also possible that the observed distribution reflects suboptimal performance of the sucrose flotation gradient under the conditions used. Despite these changes in CLN5 expression, no statistically significant differences were detected in the total protein levels of CD63 between control and G2019S LRRK2 human fibroblasts (*Figure 7—figure supplement 1B*). Collectively, these data indicate that human cells carrying pathogenic LRRK2 mutations exhibit alterations in antibody-accessible BMP comparable to those observed in MEFs, and that a pool of LRRK2 may be present in CD63/CLN5-positive endolysosomes likely enriched in BMP.

CD63-positive compartments have been implicated in exocytic processes rather than catabolic activities (*Mittelbrunn et al., 2011*; *Verweij et al., 2018*; *Verweij et al., 2022*). In line with these observations, our data indicating that a pool of LRRK2 co-fractionates with isolated CD63-positive endolysosomes prompted us to further examine the intracellular distribution of CD63 in these cells. An increased number of CD63-positive vesicles near the plasma membrane was quantified in mutant LRRK2 fibroblasts compared to control cells (*Figure 7D, E*). On average, ~25% of CD63-positive peripheral endolysosomes were also positive for BMP, with no observable differences between control and mutant LRRK2 cells (*Figure 7D, F*). The peripheral enrichment observed in G2019S LRRK2 fibroblasts suggests enhanced recruitment of CD63-positive compartments toward exocytic sites at the cell surface, supporting the notion that these vesicles may eventually fuse with the plasma membrane to release EVs, as previously proposed (*Mittelbrunn et al., 2011*; *Verweij et al., 2018*).

To determine more precisely whether LRRK2 kinase activity modulates the exocytosis of CD63-positive endolysosomes, we performed live-cell total internal reflection fluorescence (TIRF) microscopy using CD63-pHluorin, a genetically encoded fluorescent reporter of exosome release (*Lu et al., 2018*; *Verweij et al., 2018*). For these experiments, four independent control and G2019S LRRK2 patient-derived fibroblast cell lines were transduced to stably express CD63-pHluorin. The presence of BMP in CD63-pHluorin-positive endolysosomes was confirmed by immunofluorescence analysis after selection of transduced cells (*Figure 7G*). Consistent with recent findings in human induced pluripotent

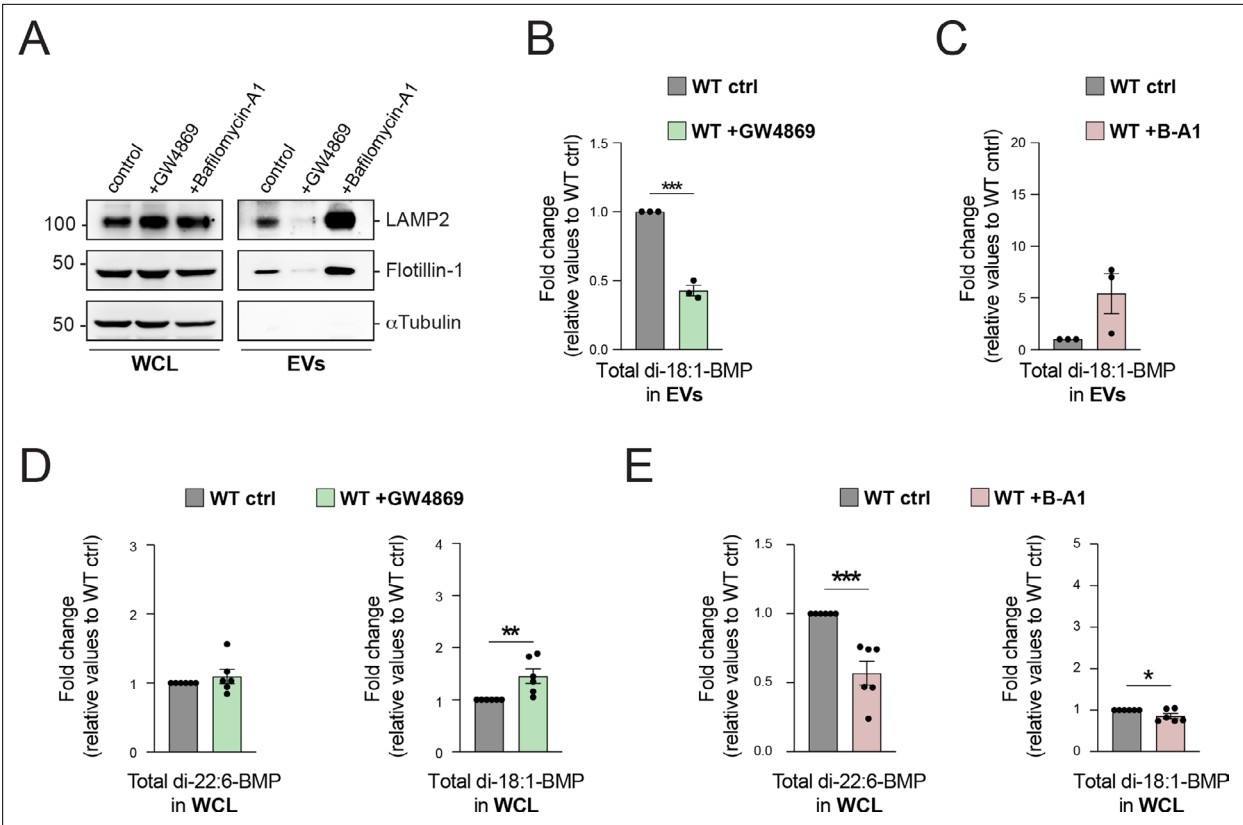

**Figure 6.** Pharmacological modulation of extracellular vesicle (EV)-mediated BMP exocytosis. (**A**) Whole cell lysates (WCL) and isolated EVs from wild-type (WT) mouse embryonic fibroblast (MEF) cells treated with 10 µM GW4869 or 10 nM bafilomycin-A1 (B-A1) for 24 hr were analyzed by immunoblotting. Representative immunoblots of LAMP2, Flotillin-1, and α-Tubulin are shown. Molecular weight marker mobility is shown in kDa. Ultra-performance liquid chromatography–tandem mass spectrometry (UPLC–MS/MS) determination of EV-associated di-18:1-BMP normalized to protein content from WT MEF cells treated with 10 µM GW4869 (**B**) or 10 nM B-A1 (**C**) for 24 hr. Data shown as fold change relative to WT control MEF cells. Only BMP isoforms that were detected are shown. Data from 6 independent experiments (mean ± SEM). Significance determined by two-tailed unpaired $t$-test; *$p < 0.05$, **$p < 0.01$, ***$p < 0.001$, ****$p < 0.0001$. Quantitation of BMP isoforms normalized to protein content in WCL from MEF WT cells treated with 10 µM GW4869 (**D**) or 10 nM B-A1 (**E**) for 24 hr. Data shown as fold change relative to WT control MEF cells. Only BMP isoforms that were detected are shown. Data from 3 independent experiments (mean ± SEM). Significance determined by two-tailed unpaired $t$-test; *$p < 0.05$, **$p < 0.01$, ***$p < 0.001$.

The online version of this article includes the following source data and figure supplement(s) for figure 6:

**Source data 1.** Uncropped blots.

**Source data 2.** Annotated uncropped blots.

**Source data 3.** Plotted values in panels B–E.

**Figure supplement 1.** Pharmacological modulation of extracellular vesicle (EV)-mediated BMP exocytosis in mutant LRRK2 mouse embryonic fibroblast (MEF) cells.

**Figure supplement 1—source data 1.** Uncropped blots.

**Figure supplement 1—source data 2.** Annotated uncropped blots.

stem cell (iPSC)-derived neurons harboring the R1441H LRRK2 pathogenic mutation (*Palumbos et al., 2025*), TIRF microscopy revealed an overall increase in endolysosomal exocytosis in the four G2019S LRRK2 fibroblast cell lines tested, which was significantly reduced upon MLi-2 treatment (*Figure 7H, I; Videos 1 and 2*). Together, these findings are consistent with our previous observations in MEFs (*Figures 2E and 3E*) and support our initial hypothesis that LRRK2 kinase activity drives exocytosis and release of BMP-enriched EVs.

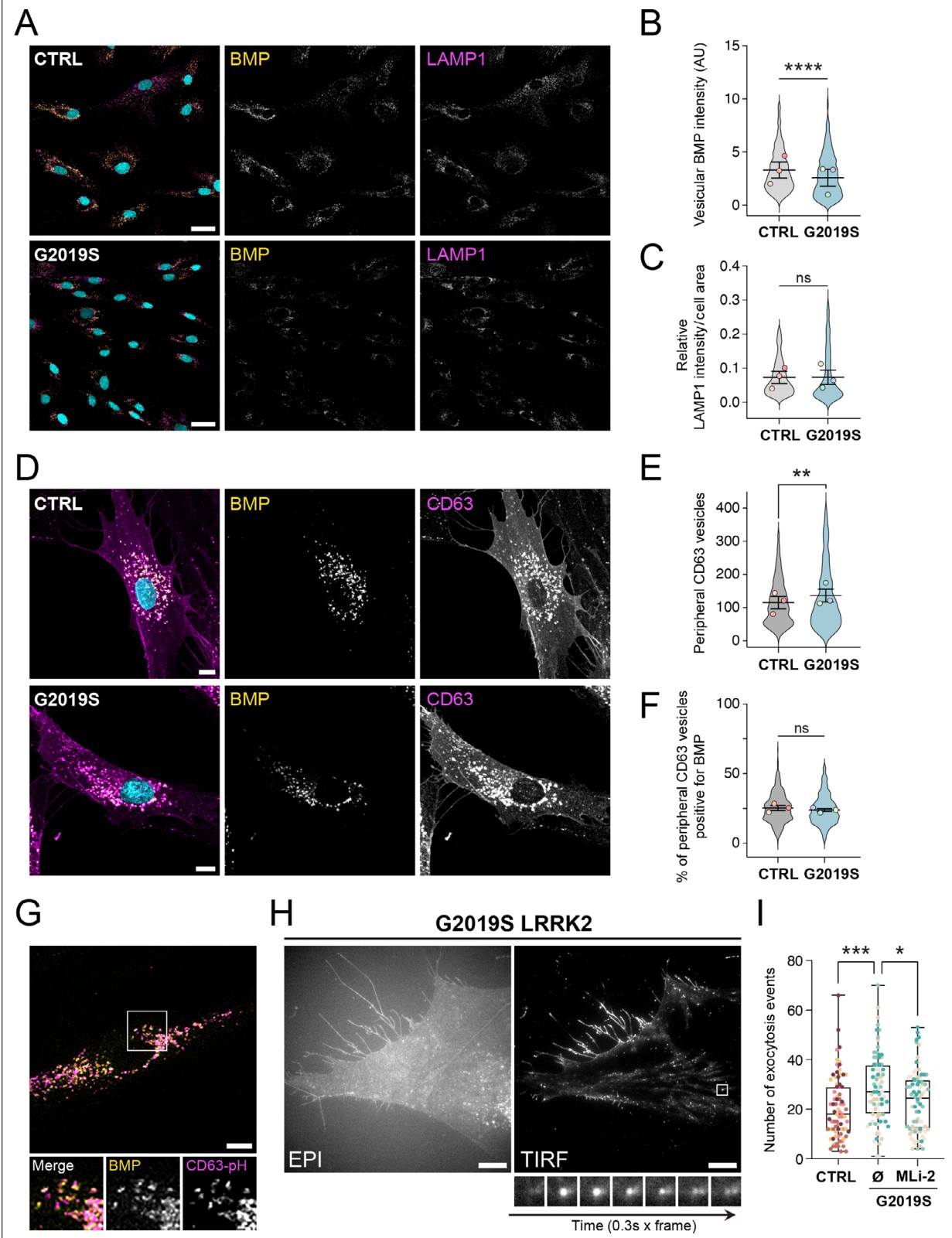

**Figure 7.** Patient-derived G2019S LRRK2 fibroblasts exhibit alterations of antibody-accessible BMP and increased endolysosomal exocytosis. (**A**) Confocal microscopy of endogenous BMP (yellow) and LAMP1 (magenta) immunofluorescence in control (CTRL) and LRRK2-G2019S mutant-derived fibroblasts. Scale bar: 40 µm. Quantification of vesicular BMP intensity (**B**) and LAMP1 relative intensity (**C**) per cell area. Colored dots represent the mean value of three independent experiments (*n* = 3 CTRL and *n* = 3 G2019S LRRK2 fibroblast cell lines); violin plots show the distribution of

*Figure 7 continued on next page*

*Figure 7 continued*

individual cell data (60 cells per independent experiment). Significance determined by two-tailed paired *t*-test; ****p < 0.0001; ns, not significant. (**D**) Confocal microscopy of endogenous BMP (yellow) and CD63 (magenta) immunofluorescence in CTRL and G2019S LRRK2 fibroblasts. Scale bar: 10 μm. Quantification of the number of CD63 vesicles in the peripheral cell region (**E**) and the percentage of peripheral CD63 vesicles positive for BMP (**F**). Colored dots represent mean values of three independent experiments (*n* = 3 CTRL and *n* = 3 G2019S LRRK2 fibroblast cell lines); violin plots show the distribution of individual cell data (60 cells per independent experiment). Significance determined by two-tailed paired *t*-test; **p < 0.01; ns, not significant. (**G**) A representative confocal microscopy image of endogenous BMP (yellow) and transduced CD63-pHluorin (magenta) in a CTRL human fibroblast cell line used for total internal reflection fluorescence (TIRF) microscopy experiments. Inset image shows co-localization between BMP and CD63-pHluorin in vesicular structures (a similar degree of co-localization was observed in G2019S LRRK2 cells; data not shown). Scale bar: 10 μm. (**H**) Representative epifluorescence (EPI) and TIRF microscopy images of the same cell from a G2019S LRRK2 patient-derived fibroblast cell line treated with vehicle (DMSO) overnight. Inset images show a TIRF microscopy time-lapse sequence (0.3 s per frame) from a single CD63-pHluorin-positive fusion event at the plasma membrane. Scale bar: 10 μm. (**I**) Quantification of CD63-pHluorin-positive fusion events in stably expressing control (CTRL) and G2019S LRRK2 fibroblasts treated with vehicle (Ø) or 200 nM MLi-2 for 16 hr. Each dot represents one cell (*n* = 20 cells quantified from four CTRL and four G2019S LRRK2 cell lines). Significance determined by Tukey's multiple comparisons test, ordinary one-way ANOVA; *p < 0.05, **p < 0.001.

The online version of this article includes the following source data and figure supplement(s) for figure 7:

**Source data 1.** IF images in panel A.

**Source data 2.** IF images in panel D.

**Source data 3.** IF images in panel G.

**Source data 4.** EPI and TIRF images in panel H.

**Source data 5.** Plotted values in panels B, C, E, F, I.

**Figure supplement 1.** Analysis of endolysosomal fractions in control and G2019S LRRK2 human-derived skin fibroblasts.

**Figure supplement 1—source data 1.** Uncropped blots.

**Figure supplement 1—source data 2.** Annotated uncropped blots.

**Figure supplement 1—source data 3.** Plotted values in panels B, C, E, F, I.

## Discussion

Here, we have shown that (a) hyperactive LRRK2 kinase upregulates mass spectrometry-determined BMP levels and the EVs that release BMP in MEF cells. (b) Pharmacological inhibition of LRRK2 and GCase using MLi-2 and CBE, respectively, modulates BMP abundance in cells and isolated exosomal fractions in opposite directions without interfering

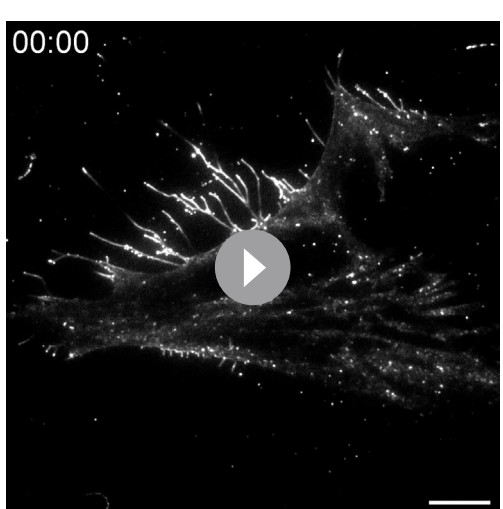

**Video 1.** Time-lapse of vehicle (DMSO)-treated human G2019S LRRK2 fibroblast cell stably expressing CD63-pHluorin. The timestamp on the upper left corner indicates seconds. Fusion events are indicated by white circles. Scale bar: 10 μm. Images were acquired at ~3.33 frames/s.

https://elifesciences.org/articles/106330/figures#video1

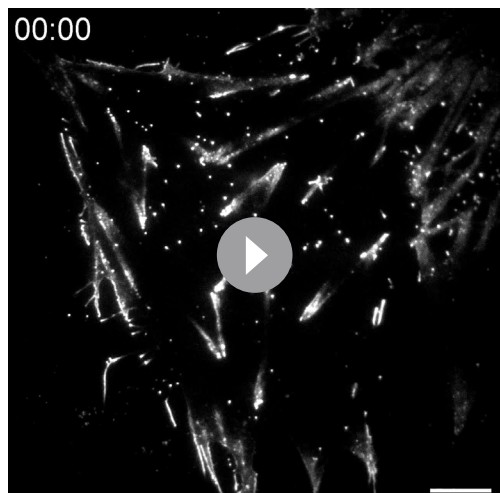

**Video 2.** Time-lapse of MLi-2-treated human G2019S LRRK2 fibroblast cell stably expressing CD63-pHluorin. Cells were treated with 200 nM MLi-2 for 16 hr prior to analysis. The timestamp on the upper left corner indicates seconds. Fusion events are indicated by white circles. Scale bar: 10 μm. Images were acquired at ~3.33 frames/s.

https://elifesciences.org/articles/106330/figures#video2

with the kinetics of BMP biosynthetic or catabolic rates, although the effect of GCase inhibition on EV-associated BMP showed more variability across experiments. While PD-associated mutations in *LRRK2* increase kinase activity, *GBA1* pathogenic variants are associated with decreased GCase enzymatic function. Therefore, MLi-2 treatment represents a potential therapeutic rescue, whereas CBE phenocopies pathogenic conditions. (c) Although the hyperactive kinase mutation is associated with an overall increase in the BMP synthesis, neither MLi-2 nor CBE affected BMP biosynthetic rate in the MEFs. (d) The expression of CLN5, a BMP synthase (*Medoh et al., 2023*; *Bulfon et al., 2024*), is upregulated in mouse and patient-derived fibroblasts with PD-associated LRRK2 mutations, consistent with the lipidomics data indicating elevated amounts of cellular BMP. (e) BMP release was modulated by pharmacological agents known to modulate EV secretion. (f) Finally, G2019S LRRK2 human fibroblasts exhibit enhanced endolysosomal exocytosis and EV release, which decreased upon MLi-2 treatment. Together, these data indicate that the previously reported increase in urinary BMP levels in LRRK2 mutation carriers reflects dysregulated exocytosis of BMP-containing EVs.

Pioneering studies by Gruenberg and colleagues underscore the importance of BMP in endolysosome homeostatic regulation and generation of endolysosomal ILVs (*Gruenberg, 2020*; *Kobayashi et al., 1998*; *Matsuo et al., 2004*) that become exosomes upon release. Subsequent studies revealed that cellular levels of this atypical phospholipid are invariably altered in many neurodegenerative disorders characterized by endolysosome dysfunction (*Akgoc et al., 2015*). Recently, key lysosomal enzymes involved in distinct steps of the BMP biosynthetic pathway were identified, including CLN5, which plays a critical role in this process (*Medoh et al., 2023*; *Bulfon et al., 2024*). Our microscopy analysis revealed decreased antibody-accessible BMP levels in R1441G LRRK2 cells. In contrast, our targeted lipid pathway analysis measurements consistently showed higher total BMP levels in R1441G LRRK2 cells compared with WT cells. As we have reported previously (*Lu et al., 2022*), BMP antibody detects only a sub-pool of 'accessible' BMP while mass spectrometry detects all pools. Alternately, not all BMP isoforms may be detected equally well. Given the essential roles of BMP in endolysosomal catabolism, our immunofluorescence data predict that LRRK2 mutant cells have defective degradative capacity, consistent with recent reports (*Henry et al., 2015*; *Yadavalli and Ferguson, 2023*). In addition, in both MEFs and patient-derived mutant LRRK2 fibroblasts, we found upregulation of the BMP-synthesizing enzyme CLN5 and, in MEFs, a concomitant increase in LAMP2 protein levels. These findings suggest that LRRK2 may participate in endolysosome biogenesis regulation. Indeed, in macrophages and microglia, LRRK2 regulates the levels of multiple lysosomal proteins by inhibiting TFEB and MiTF (*Yadavalli and Ferguson, 2023*), transcription factors that activate lysosome-related gene expression by binding to coordinated lysosomal expression and regulation (CLEAR) elements (*Sardiello et al., 2009*). Although the promoter region of CLN5 contains a potential TFEB binding site (*Sardiello et al., 2009*), formal evidence for TFEB-mediated transcription of CLN5 is still lacking. The increase in CLN5 (and LAMP2 in MEFs) protein expression may also reflect lysosomal stress (*Sardiello et al., 2009*). Interestingly, MLi2 restored CLN5 levels in both murine-derived and human fibroblasts, further providing support that BMP is a pharmacodynamic biomarker of not just target modulation but also a potential disease-relevant lysosomal pathway modulation. Our findings of CLN5 upregulation in mutant LRRK2 MEFs, validated by similar results in patient-derived fibroblasts, further establish MEFs as a relevant cellular model system for studying LRRK2 kinase activity and its therapeutic targeting (*Dhekne et al., 2018*; *Sobu et al., 2021*; *Dhekne et al., 2021*; *Kania et al., 2023*). Taken together, our data indicate that LRRK2 activity modulates specific components of the lysosomal network, including CLN5 and LAMP2. Nevertheless, quantitative analysis of additional endolysosomal markers (LAMP1 and CD63) in human fibroblasts did not reveal statistically significant differences between control and mutant LRRK2 cells. The elevated LAMP2 expression observed in the engineered MEF clone expressing R1441G may reflect a cell type-specific effect, potentially linked to differential penetrance of LRRK2 signaling on the lysosome biogenesis response.

We and others reported that BMP is aberrantly high in urine derived from LRRK2 and GBA mutation carriers (*Alcalay et al., 2020*; *Merchant et al., 2023*; *Gomes et al., 2023*). These results have also been reproduced in animal models, in which urinary BMP levels decrease upon administration of MLi-2 and other LRRK2 kinase inhibitors (*Fuji et al., 2015*; *Baptista et al., 2020*; *Jennings et al., 2022*). The present study complements these previous observations by providing evidence that EV-mediated BMP release can be regulated by LRRK2 and, although less consistently, by GCase enzymatic activity. Consistent with this model, recent studies have shown that aberrant LRRK2 and GCase activities

influence EV secretion in cellular and animal models of PD, including human patients (*Papadopoulos et al., 2018*; *Cerri et al., 2021*; *Maloney et al., 2025*). Moreover, our data showing reduced exocytosis and EV release in mutant LRRK2 MEFs and patient-derived fibroblasts treated with MLi-2 is consistent with previous observations that LRRK2 kinase inhibition leads to surfactant accumulation in type 2 pneumocytes in the lung, likely as a consequence of impaired lysosome-related organelle exocytosis (*Baptista et al., 2020*). Moreover, a recent study also reported decreased exocytosis of CD63-pHluorin-positive compartments in MLi-2-treated primary murine neurons and human iPSC-derived neurons harboring pathogenic LRRK2 mutations (*Palumbos et al., 2025*) further supports our conclusions.

BMP is enriched in urinary EVs (*Rabia et al., 2020*). However, other studies did not detect BMP enrichment in isolated EV fractions and concluded that BMP may be present in ILVs of endolysosomal subpopulations devoted to degradative activities rather than exocytic events (*Wubbolts et al., 2003*; *Laulagnier et al., 2004*), as previously proposed (*White et al., 2006*). Using our UPLC–MS/MS methodology, extracellular BMP can be efficiently detected in the urine of healthy and mutant LRRK2/GCase carriers (*Alcalay et al., 2020*; *Merchant et al., 2023*; *Gomes et al., 2023*), and additionally, in EV fractions derived from both WT and R1441G LRRK2 MEFs (this study). How does LRRK2 influence the release of BMP-positive EVs (BMP-EVs)? A subset of Rab proteins that are master regulators of intracellular membrane trafficking pathways is important LRRK2 substrates (*Pfeffer, 2023*). Some of these Rabs include Rab10, Rab12, and Rab35 and have been shown to play regulatory roles in endolysosomal exocytosis and EV release (*Hsu et al., 2010*; *Vieira, 2018*). We thus speculate that phosphorylation of one or several of these Rabs may lead to enhanced BMP release. Indeed, G2019S LRRK2-mediated Rab35 phosphorylation has been proposed to promote EV-mediated α-synuclein release and propagation between cells (*Bae et al., 2018*; *Bae and Lee, 2020*). It is likely that aberrant LRRK2 and GCase activities trigger exocytosis and associated release of BMP-containing EVs via a so-called clearance pathway activated due to accumulation of cytotoxic endolysosomal substrates (*Settembre et al., 2013*; *Tsunemi et al., 2019*). The contribution of these phosphorylated Rabs to EV-mediated BMP release will be of interest to future work.

In summary, this work provides evidence that pathological LRRK2 activity, and to a more variable degree, GCase dysfunction, enhance EV-mediated BMP release without altering its metabolism in cells. The observed changes induced by MLi-2 highlight the potential of LRRK2 inhibition as a therapeutic strategy to restore endolysosomal homeostasis by reducing BMP production and aberrant EV release. What is the clinical significance of elevated urinary BMP levels in PD patients? Inhibition of LRRK2 improves endolysosomal function (*Jennings et al., 2022*), consistent with increased LRRK2 and decreased GCase activities associated with pathogenic variants which worsen endolysosomal homeostasis (*Roosen and Cookson, 2016*; *Do et al., 2019*); this may eventually lead to endolysosomal exocytosis followed by EV-mediated BMP release. We speculate that BMP-EVs may harbor a distinct molecular repertoire that could perhaps inform more precisely disease progression or pathobiology. Future investigations will be needed to study the contribution of dysfunctional LRRK2 and GCase in exocytosis of specific BMP-positive endolysosomal subpopulations and to further characterize the role of BMP-EVs in the context of disease pathophysiology but also as a PD diagnostic tool.

## Materials and methods

**Key resources table**

| Reagent type (species) or resource | Designation | Source or reference | Identifiers | Additional information |
|---|---|---|---|---|
| Cell line (*Mus musculus*) | WT and R1441G LRRK2 MEFs | https://mrcppureagents.dundee.ac.uk/ | | |
| Cell line (*Homo sapiens*) | Control and G2019S LRRK2 fibroblasts | *Fernández-Santiago et al., 2021* | | |
| Recombinant DNA reagent | CD63-pHluorin | *Lu et al., 2018* | | |
| Antibody | anti-mouse LAMP2, clone GL2A7 (Rat monoclonal) | Developmental Studies Hybridoma Bank | GL2A7; RRID:AB_2281134 | IF (1:500) WB (1:1000) |

*Continued on next page*

*Continued*

| Reagent type (species) or resource | Designation | Source or reference | Identifiers | Additional information |
|---|---|---|---|---|
| Antibody | anti-Flotillin-1 (mouse monoclonal) | BD Biosciences | 610821; RRID:AB_398140 | WB (1:1000) |
| Antibody | anti-phospho-Rab10 (phospho T73) (Rabbit monoclonal) | Abcam | ab230261; RRID:AB_2811274 | WB (1:1000) |
| Antibody | Anti-alpha tubulin (mouse monoclonal) | Sigma-Aldrich | T5168; RRID:AB_477579 | WB (1:10000) |
| Antibody | anti-CLN5 (Rabbit monoclonal) | Abcam | ab170899; RRID:AB_3662651 | WB (1:1000) |
| Antibody | anti-phosphoLRRK2 (phosphor S935) (Rabbit monoclonal) | Abcam | ab133450; RRID:AB_2732035 | WB (1:1000) |
| Antibody | anti-LAMP1 XP D2D11 (Rabbit monoclonal) | Cell Signalling | 9091; RRID:AB_2687579 | IF (1:1000) |
| antibody | anti-CD63 [EPR22458-280] (Rabbit monoclonal) | Abcam | ab252919 | IF (1:500) WB (1:1000) |
| Antibody | anti-LRRK2 (Rabbit monoclonal) | Abcam | ab133474; RRID:AB_2713963 | WB (1:500) |
| Antibody | anti-calnexin (mouse monoclonal) | BD Transduction | 610523 | WB (1:1000) |
| Antibody | anti-GFP (Rabbit polyclonal) | Abcam | ab290; RRID:AB_303395 | WB (1:500) |
| Chemical compound, drug | MLi-2 | Tocris Bioscience | 5756 | 200 nM |
| Chemical compound, drug | Conduritol β-epoxide | MERCK | 234599 | 300 μM |
| Chemical compound, drug | Bafilomycin-A1 | Sigma-Aldrich | B1793 | 10 nM |
| Chemical compound, drug | GW4869 | SelleckChem | S7609 | 10 μM |
| Software, algorithm | FIJI | http://fiji.sc/ | SCR_002285 | |
| Software, algorithm | CellProfiler | https://cellprofiler.org/ | SCR_007358 | |

## Cell culture, antibodies, and other reagents

None of the cell lines used in this study are listed in the International Cell Line Authentication Committee database of commonly misidentified cell lines. Human-derived skin fibroblasts were obtained as previously described (*Fernández-Santiago et al., 2021*). WT and R1441G LRRK2 MEF cells were kindly provided by Dr. Dario Alessi (MRC-PPU, University of Dundee) and generated as previously described (*Ito et al., 2016*). All cell lines in this study were grown in Dulbecco's modified Eagle's media (DMEM) containing 10% fetal bovine serum, 2 mM L-glutamine, and penicillin (100 U/ml)/streptomycin (100 mg/ml). Cell lines were cultured at 37°C with 5% $CO_2$. All cells were regularly tested for Mycoplasma PCR products. EV-free media was prepared by overnight ultracentrifugation of DMEM or RPMI supplemented with 10% FBS at 100,000 × *g* in a SW32Ti rotor. For bafilomy-cin-A1 (Sigma-Aldrich; catalog no. B1793) and GW4869 (SelleckChem; catalog no. S7609), the drug was added to cell media at 10 nM and 10 μM, respectively, and cells were cultured for 24 hr before exosome collection. For MLi-2 (Tocris Bioscience; catalog no. 5756) and CBE (MERK; catalog no. 234599), the drugs were added to cell media at 200 nM and 300 μM, respectively. In our experiments, 48-hr incubations were necessary to sustain full LRRK2/GCase inhibition throughout the EV collection period. EV biogenesis, BMP biosynthesis, and packaging into EVs are time-dependent processes; therefore, extended incubation and collection periods (≥48 hr) were required to allow downstream effects of LRRK2/GCase inhibition on BMP production and release to manifest, and to obtain sufficient EV material for biochemical and lipidomic analyses. 0.167 mM Oleic acid (Cambridge Isotope Laboratories; catalog no. DLM-10012-0.001) and Docosahexaenoic acid (Cambridge Isotope Laboratories; catalog no. DLM-10012-0.001) were conjugated with 0.0278 mM fatty acid-free BSA (Sigma-Aldrich;

A8806) at a 1:6 molar ratio. Primary antibodies diluted in phosphate-buffered saline (PBS) with 1% BSA (for immunofluorescence) or 5% skim milk (for immunoblotting) were mouse anti-lysobisphosphatidic acid (anti-BMP) clone 6C4 1:1000 (EMD Millipore; catalog no. MABT837), rat monoclonal anti-mouse LAMP2 1:500/1:1000 (Developmental Studies Hybridoma Bank; catalog no. GL2A7), mouse anti-Flotillin-1 1:1000 (BD Biosciences; catalog no. 610821), rabbit anti-phospho-Rab10 1:1000 (Abcam; catalog no. ab230261), mouse anti-α-Tubulin 1:10,000 (Sigma-Aldrich; catalog no. T5168), rabbit anti-CLN5 1:000 (Abcam; catalog no. ab170899), rabbit anti-phosphoLRRK2 1:000 (Abcam; catalog no. ab133450), rabbit anti-LAMP1 XP D2D11 1:1000 (Cell Signalling; catalog no. 9091), rabbit anti-CD63 [EPR22458-280] 1:500/1:1000 (Abcam; catalog no. AB252919), rabbit anti-LRRK2 1:500 (Abcam; catalog no. AB133474), mouse anti-calnexin 1:1000 (BD Transduction; 610523), and rabbit anti-GFP 1:500 (Abcam; catalog no. AB290).

## Immunofluorescence

WT and R1441G LRRK2 mutant MEF cells were fixed with 3.7% (vol/vol) paraformaldehyde for 15 min, washed, and permeabilized/blocked with 0.1% saponin/1% BSA-PBS before staining with anti-LAMP2 and anti-BMP antibodies followed by goat anti-rat AlexaFluor555 and donkey anti-mouse AlexaFluor488. Nuclei were stained using 0.1 mg/ml DAPI (Sigma-Aldrich; catalog no. D9542). Coverslips were mounted on glass slides with Mowiol. Microscopy images were acquired using a Zeiss LSM880 laser scanning spectral confocal microscope (Carl Zeiss) equipped with an Axio Observer 7 inverted microscope, a blue diode (405 nm), argon (488 nm), diode-pumped solid-state (561 nm), and HeNe (633 nm) lasers, and a Plan Apochromat 63×/NA1.4 oil-immersion objective lens. DAPI, Alexa Fluor 488, and Alexa Fluor 555 were acquired sequentially using 405-, 488-, and 561 nm laser lines; Acousto-Optical Beam Splitter as a beam splitter; and emission detection ranges 415–480, 500–550, and 571–625 nm, respectively. The confocal pinhole was set at 1 Airy unit. All images were acquired in a 1024 × 1024-pixel format. BMP and LAMP2 intensities were quantified using CellProfiler (*Carpenter et al., 2006*). All images for BMP-LAMP1 and BMP-CD63 analyses in human fibroblasts were acquired in tile-scan mode (5 × 5 field of view) with a z-step of 5, yielding a 2354 × 2354-pixel format. Tile-scan images were stitched with a 10% overlap, and maximum-intensity projections were generated using the Zeiss Zen Blue software. BMP and LAMP1 fluorescence intensities, as well as the number of peripheral CD63-positive vesicles and the percentage of peripheral CD63 vesicles positive for BMP, were quantified using Fiji (ImageJ) software.

## Lentivirus production and cell transduction

To produce CD63-pHluorin lentivirus, a lentivirus vector containing the CD63-pHluorin-expressing cassette (*Lu et al., 2018*) was combined with third-generation lentiviral packaging mix (1:1:1 mix of pMD2.G, pRSV-Rev, and pMDLg/pRRE) at a 1:1 ratio and transfected into HEK293T cells using polyethylenimine (PEI, #23966-2, PolySciences Inc), as previously described (*Dhekne et al., 2023*). Transfected cells were grown for 72 hr and then the supernatant containing the viral particles was collected and passed through a sterile 0.45-µm Millex syringe filter (Millipore). Human-derived skin fibroblasts grown in 60 mm culture dishes were infected by adding a 1:1 mixture of complete DMEM medium and viral supernatant supplemented with 8 mg/ml polybrene, followed by a 24-hr incubation. The supernatant was removed, and cells were grown in fresh DMEM medium for 48–72 hr. Infected cells were selected by growing in DMEM medium supplemented with 1–2 mg/ml puromycin (Sigma-Aldrich) for 3–5 days. Cells stably expressing CD63-pHluorin were recovered in complete DMEM medium lacking puromycin for 48–72 hr.

## TIRF live microscopy

Control and G2019S LRRK2 human fibroblasts stably expressing CD63-pHluorin were cultured on 25 mm coverslips and treated with DMSO (vehicle) or 200 nM MLi-2. After 16 hr, cells were imaged using a Zeiss LSM880 Airyscan Elyra PS.1 laser scanning spectral confocal microscope (Carl Zeiss) equipped with an Axio Observer Andor1 inverted microscope. Imaging was performed with an α Plan-Apochromat TIRF 100×/1.46 NA oil DIC objective, a BP495–575 + LP750 filter set, and an sCMOS camera operated in TIRF mode. Excitation was achieved with an HR diode laser (488 nm, 200 mW). Time-lapse videos were acquired at 0.30 s per frame for a total of 540 frames per cell analyzed. All imaging experiments were performed at 37°C in a humidified incubator (5% $CO_2$). Fusion events for

each cell were detected and quantified as sudden increases in fluorescence at the cell surface using the AMvBE macro in Fiji (ImageJ), as described (*Bebelman et al., 2020*). Twenty cells were imaged and quantified for each of the four fibroblast cell lines within each cohort.

## Immunoblotting

Cells were lysed in lysis buffer (50 mM HEPES, 150 mM KCl, 1% Triton X-100, 5 mM $MgCl_2$, pH 7.4) supplemented with a protease/phosphatase inhibitor cocktail (1 mM $Na_3VO_4$, 10 mM NaF, 1 mM PMSF, 10 µg/ml leupeptin, and 10 µg/ml aprotinin). After centrifugation at 12,000 × *g* for 10 min, protein concentrations were measured in the cleared lysates. Equal protein amounts were loaded in lanes corresponding to WCL fractions and normalized based on α-Tubulin levels. For the EV fractions, all protein recovered from EV pellets after isolation was loaded. Exosome final pellets were resuspended in SDS–PAGE loading dye (50 mM Tris-HCl pH 6.8, 6% glycerol, 2% SDS, 0.2% bromophenol blue), heated briefly at 95°C, resolved by 10% SDS–PAGE gel and transferred onto nitrocellulose membranes (Bio-Rad Laboratories; catalog no. 1620115) using a Bio-Rad Trans-Blot system. Membranes were blocked with 5% skim milk in TBS with Tween-20 for 60 min at RT. Primary antibodies were diluted in blocking buffer and incubated overnight at 4°C. HRP-conjugated secondary antibodies (Bio-Rad Laboratories) diluted in blocking buffer at 1:3000 were incubated for 60 min at RT and developed using EZ-ECL (Biological Industries) or SuperSignal West Femto (Thermo Scientific; catalog no. 34094). Blots were imaged using an ImageQuant LAS 4000 system (GE Healthcare) and quantified using ImageJ software.

## Transmission electron microscopy

Cells in culture were washed in PBS and fixed for 1 hr in 2.5% glutaraldehyde in 0.1 M phosphate buffer (PB) at RT. Next, samples were slowly and gently scraped and pelleted in 1.5 ml tubes. Pellets were washed in PB and incubated with 1% $OsO_4$ for 90 min at 4°C. Then samples were dehydrated, embedded in Spurr, and sectioned using Leica ultramicrotome (Leica Microsystems). Ultrathin sections (50–70 nm) were stained with 2% uranyl acetate for 10 min, a lead-staining solution for 5 min and observed using a transmission electron microscope, JEOL JEM-1010 fitted with a Gatan Orius SC1000 (model 832) digital camera. MVE and ILVs were identified by morphology and MVE area and ILV number were measured using ImageJ.

## EV isolation

MEF adherent cell cultures were seeded at 3–4 × $10^6$ cells/plate in EV-free complete DMEM and grown for 24–48 hr. Cell cultures were centrifuged at 300 × *g*, 5 min to pellet cells, and supernatants were centrifuged again at 2100 × *g*, 20 min to pellet dead cells. After this, filtration through a 0.22-µm filter unit (Millipore) was performed. Filtered media were ultracentrifuged for 75 min at 100,000 × *g* in SW32Ti rotor to pellet EVs. The supernatant was discarded, and the pellet (small EVs) was re-suspended in PBS and centrifuged again for 75 min at 100,000 × *g* in 140AT rotor. Exosome pellets were frozen and stored at –80°C prior to further processing. In all EV-related experiments, we seeded the same number of EV-producing cells per condition, and the resulting EV-derived data (from both immunoblotting and lipidomics analyses) were normalized to the corresponding WCL protein content to ensure comparability across conditions.

## Lipidomic analysis

Targeted quantitative ultra-performance liquid chromatography tandem mass spectrometry was used to accurately quantitate the three geometrical isomers (2,2-, 2,3- and 3,3-) of di-22:6-BMP and di-18:1-BMP in control or treated MEF cells. Lipidomic analyses were conducted by Nextcea, Inc as previously described (*Hsu et al., 2010*). Standard curves were prepared using authentic BMP reference standards. Protein was determined by bicinchoninic acid protein assay. A multiplexed quantitative ultraperformance liquid chromatography tandem mass spectrometry method was used to simultaneously quantitate sample glucosylceramides (GluCer d18:1/16:0, d18:1/18:0, d18:1/22:0, d18:1/24:0, d18:1/24:1), galactosylceramides (GalCer d18:1/16:0, d18:1/18:0, d18:1/22:0, d18:1/24:0, d18:1/24:1), glucosylsphingosine (GluSph 18:1), and galactosylsphingosine (GalSph 18:1). Standard curves were prepared from related standards using a class-based approach. Internal standards were used for each analyte reported. A SCIEX Triple Quad 7500 mass spectrometer was used in positive

electrospray ionization mode for detection (SCIEX, Framingham, MA, USA). Injections were made using a Shimadzu Nexera XR UPLC (ultraperformance liquid chromatography) system (Shimadzu Scientific Instruments, Kyoto, Japan). The instruments were controlled by SCIEX OS 2.0 software.

## Metabolic labeling

WT and R1441G LRRK2 mutant MEF cells were treated with 200 nM MLi-2 or 300 µM CBE for 16 hr. Cells were washed twice with PBS and incubated in serum-free DMEM ± MLi-2 or CBE for 1 hr. Next, incubate cells in 0.167 mM $^{13}$C-labeled oleic acid (OA) and 0.167 mM deuterated docosahexaenoic acid in serum-free DMEM containing 0.0278 mM BSA for 25 min. Cells were washed with serum-free DMEM and chased with serum-free DMEM for short time points (0, 15, 30, 45, and 60 min) or complete DMEM for long time points (8, 24, and 48 hr). Cells were washed with PBS, trypsinized, and collected in cold PBS. Cells were pelleted by centrifugation at 1000 rpm for 7 min and stored at –80°C prior to further processing.

## GCase activity

GCase activity was measured by using the fluorogenic substrate PFB-FDGlu (Invitrogen, Carlsbad, CA). WT and R1441G LRRK2 mutant MEF cells were treated with 300 µM CBE for 24 hr. After this, 150 µg/ml PFB-FDGlu (Thermo Fisher) was then added to cells followed by incubation for 2 hr at 37°C. Cells were then washed twice with PBS, trypsinized, and fixed in cold 2% paraformaldehyde–PBS for 30 min. Cells were washed twice in PBS before assayed on an FACS analyzer (FACScan; BD) equipped with a 488-nm laser and 530-nm FITC filter. Data were analyzed using Flowjo software (Tree Star).

## Subcellular fractionation

Endolysosomes were isolated by ultracentrifugation of discontinuous sucrose density gradients as previously described (*Meneses-Salas et al., 2020*). Briefly, control and G2019S LRRK2 fibroblasts were collected using a homogenization buffer containing 3 mM Imidazole, 250 nM sucrose, pH 7.4, and protease inhibitors. Cell pellets were homogenized by 90 passages through a 27-gauge needle. Complete homogenization was assessed by phase-contrast microscopy. After this, samples were centrifuged for 10 min at 1000 × *g*. The post-nuclear supernatant (PNS) fractions were collected and adjusted to 40,2% (wt/vol) sucrose. PNS fractions were then loaded at the bottom of a micro-ultracentrifuge tube for Himac P70AT(RP70AT) Fixed Angle Rotor. Then, 35% sucrose, 25% sucrose, and 8% sucrose layers were poured stepwise over the PNS bottom layer. The discontinuous sucrose gradient was centrifuged for 3 hr at 100,000 × *g* using a Himac CS-(F)NX (Eppendorf) micro-ultracentrifuge. After this, 15 fractions were collected, and protein was trichloroacetic acid-precipitated before SDS–PAGE analysis.

## Statistics

Results are expressed as the mean ± SEM. Means were compared using Student's *t*-test when two experimental conditions were independently compared. Unless otherwise noted, statistical significance between multiple comparisons was assessed via one- or two-way ANOVA with uncorrected Fisher's LSD test using GraphPad Prism version 9.

## Acknowledgements

This research was funded by a grant to KM and AL from The Michael J Fox Foundation for Parkinson's Research (MJFF-019043; MJFF-023914). RFS was supported by a Miguel Servet grant (#CP19/00048), two FIS grants (#PI20/00659 and #PI23/00661), a PFIS grant (#FI21/00104), and a M-AES grant (#MV22/00041) from the Instituto de Salud Carlos III (ISCIII) co-funded by the European Union, and a "Consolidación Investigadora" grant (#CNS2025-166810) from the Agencia Estatal de Investigación (Spain). IDIBAPS receives support from the CERCA program of Generalitat de Catalunya.

## Additional information

### Competing interests

Marianna Arnold, Frank Hsieh: Employed by Nextcea, Inc, which holds patent rights to the di-22:6-BMP and 2,2'-di-22:6-BMP biomarkers for neurological diseases involving lysosomal dysfunction (US 8,313,949, Japan 5,702,363, and Europe EP2419742). Suzanne R Pfeffer, Albert Lu: Has received research funding from the Michael J. Fox Foundation for Parkinson's Research. Kalpana Merchant: Has consulted for AcureX, Caraway, Nitrase, Nura Bio, Retromer Therapeutics, Sinopia Biosciences, Vanqua, Vida Ventures and the Michael J. Fox Foundation for Parkinson's Research; and has received research funding from the Michael J. Fox Foundation for Parkinson's Research. The other authors declare that no competing interests exist.

## Funding

| Funder | Grant reference number | Author |
| --- | --- | --- |
| Michael J. Fox Foundation | MJFF-019043 | Kalpana Merchant |
| Michael J. Fox Foundation | MJFF-023914 | Albert Lu |

The funders had no role in study design, data collection, and interpretation, or the decision to submit the work for publication.

## Author contributions

Elsa Meneses-Salas, Data curation, Formal analysis, Writing – review and editing; Moises Castellá, Writing – review and editing, Performed TIRF microscopy assays and subcellular fractionation experiments; Marianna Arnold, Data curation, Formal analysis, Methodology, Performed targeted lipidomics analysis; Frank Hsieh, Conceptualization, Resources, Data curation, Formal analysis, Methodology, Writing – review and editing, Supervised all lipidomics analyses; Rubén Fernández-Santiago, Provided human-derived fibroblast cell lines and participated in discussions; Mario Ezquerra, Provided human-derived fibroblast cell lines and participated in discussions; Alicia Garrido, Provided human-derived fibroblast cell lines; María-José Martí, Provided human-derived fibroblast cell lines; Carlos Enrich, Writing – review and editing, Electron microscopy analysis; Suzanne R Pfeffer, Conceptualization, Data curation, Formal analysis, Investigation, Methodology, Writing – review and editing; Kalpana Merchant, Conceptualization, Supervision, Funding acquisition, Project administration, Writing – review and editing; Albert Lu, Conceptualization, Resources, Data curation, Formal analysis, Supervision, Funding acquisition, Validation, Investigation, Visualization, Methodology, Writing – original draft, Project administration

## Author ORCIDs

Elsa Meneses-Salas https://orcid.org/0000-0001-7550-253X
Moises Castellá https://orcid.org/0000-0002-5418-4242
Marianna Arnold https://orcid.org/0009-0006-1743-4835
Frank Hsieh https://orcid.org/0000-0001-5203-8922
Rubén Fernández-Santiago https://orcid.org/0000-0002-4582-0702
Mario Ezquerra https://orcid.org/0000-0003-3246-6641
Alicia Garrido https://orcid.org/0000-0003-0652-6348
María-José Martí https://orcid.org/0000-0002-3874-967X
Carlos Enrich https://orcid.org/0000-0003-0382-2993
Suzanne R Pfeffer https://orcid.org/0000-0002-6462-984X
Kalpana Merchant https://orcid.org/0000-0003-1118-9403
Albert Lu https://orcid.org/0000-0002-7507-3330

Reviewer #1 (Public review): https://doi.org/10.7554/eLife.106330.3.sa1
Reviewer #2 (Public review): https://doi.org/10.7554/eLife.106330.3.sa2
Author response https://doi.org/10.7554/eLife.106330.3.sa3

# Additional files

## Supplementary files

MDAR checklist

## Data availability

The source data attached in our manuscript contain all the numerical data used to generate the corresponding figures. Source data files have been provided for all figures.

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
