## [Editor Report · eLife Assessment]

This **useful** study presents the potentially interesting idea that LRRK2 regulates cellular BMP levels and their release via extracellular vesicles, with GCase activity further modulating this process in mutant LRRK2-expressing cells. However, some of the evidence supporting these conclusions remains **incomplete**, and additional work is suggested under certain conditions. Overall, the study will be of interest to cell biologists working on Parkinson's disease.

---

## [Referee Report · Reviewer #1 (Public review)]

Summary:

Even though mutations in LRRK2 and GBA1 (which encodes the protein GCase) increase the risk of developing Parkinson's disease (PD), the specific mechanisms driving neurodegeneration remain unclear. Given their known roles in lysosomal function, the authors investigate how LRRK2 and GCase activity influence the exocytosis of the lysosomal lipid BMP via extracellular vesicles (EVs). They use fibroblasts carrying the PD-associated LRRK2-R1441G mutation and pharmacologically modulate LRRK2 and GCase activity.

Strengths:

The authors examine both proteins at endogenous levels, using MEFs instead of cancer cells. The study's scope is potentially interesting and could yield relevant insights into PD disease mechanisms.

Weaknesses:

Many of the authors' conclusions are overstated and not sufficiently supported by the data. Several statistical errors undermine their claims. Pharmacological treatment is very long, leading to potential off target effects. Additionally, the authors should be more rigorous when using EV markers.

Comments on revisions:

The authors have not addressed most of my concerns. For example, instead of trying with a 1-2 hour MLi2 treatment, they cited all the papers that use extremely long time points for LRRK2 inhibition; the fact that other groups do it does not mean it is biologically correct. They also refused to quantify their western blots in a proper manner, without the "hyper-normalization" claiming that it is an accepted way to quantify western blots. Again, it is statistically incorrect and biologically impossible. They also do not have a satisfactory explanation as to why the R1441G cells (which increase LRRK2 kinase activity) have no effect on EV release, but they still claim it is LRRK2 kinase activity dependent.

Overall, I am very confused by the model proposed by the authors. They only see increased EV release in the G2019S expressing cells, but not the R1441G cells, yet they claim that the increase of EV release is LRRK2 kinase activity dependent. Then, they claim that the presence of BMP (unchanged in R1441G vs CTL) in EVs is also LRRK2 kinase activity dependent. Finally, they perform TIRF with pHluorin-CD63 construct and observed an increase in G2019S cells vs CTL "further confirming that BMP release is associated with EV secretion". First, I could not see the increase in BMP release in G2019S cells (if I missed it, I apologize). And second, why didn't they do this experiment in R1441G cells? As, the R1441G cells have not displayed an increase in EV release compared to CTL cells, it could also be possible that the BMP release might be more abundant through lysosomal exocytosis (which could explain the pHluorin results) than EVs. Overall, the authors nicely demonstrate that the R1441G cells have more BMP species, likely due to increase CLN5 expression, but the release of the BMP is still not clear to this reviewer.

---

## [Referee Report · Reviewer #2 (Public review)]

Summary:

In this paper, authors used MEFs expressing the R1441G mutant of leucine-rich repeat kinase 2 (LRRK2), a mutant associated with the early onset of Parkinson's disease. They report that in these cells LAMP2 fluorescence is higher but BMP fluorescence is lower, MVE size is reduced and that MVEs contain less ILVs. They also report that LAMP2-positive EVs are increased in mutant cells in a process sensitive to LRRK2 kinase inhibition but are further increased by glucocerebrosidase (GCase) inhibition, and that total di-22:6-BMP and total di-18:1-BMP are increased in mutant LRRK2 MEFs compared to WT cells by mass spectrometry. They also report that LRRK2 kinase inhibition partially restores cellular BMP levels, and that GCase inhibition further increased BMP levels, and that in EVs from the LRRK2 mutant, LRRK2 inhibition decreases BMP while GCase inhibition has the opposite effect. Moreover, they report that BMP increase is not due to increased BMP synthesis, although authors observe that CLN5 is increased in LRRK2 mutant cells. Finally, they report that GW4869 decreases EV release and exosomal BMP, while bafilomycin A1 increases EV release. They conclude that LRRK2 regulates BMP levels (in cells) and release (via EVs). They also conclude that the process is modulated by GCase in LRRK2 mutant cells, and that these studies may contribute to the use of BMP-positive EVs as a biomarker for Parkinson's disease and associated treatments.

Strengths:

This is a potentially interesting paper,. However, I had comments that authors needed to address to clarify some aspects of their study.

Weaknesses:

(1) The authors seem to have missed the point in their reply to my first comment. They mention the paper by Stuffers et al., who reports that endosome biogenesis continues without ESCRT. This is a nice paper, but it is irrelevant to the subject at hand. In my initial comment, I drew the author's attention to an apparent contradiction: higher LAMP2 staining in R1441G LRRK2 knock-in MEFs and yet smaller MVEs with a reduced surface area. LAMP2 being one of the major glycoproteins of MVE's limiting membrane, one would have expected lower LAMP2 staining if cells contain fewer and smaller MVEs. Authors now state that elevated LAMP2 expression in cells expressing R1441G reflects a cell type-specific effect (differential penetrance of LRRK2 signaling on lysosomal biogenesis), because amounts of LAMP1 and CD63 are similar in cells from LRRK2 G2019S PD patients and control cells (new Fig 7A-F). However, authors still conclude that LRRK2 modulates the lysosomal network, including LAMP2 and CLN5. Does it?

Similarly, the mass spec analysis of BMP (Fig S1H) does not support the data in Fig 1. Does this Table include all major isoforms found in these cells? If so, the dominant isoform is by far the di-18:1 isoform in wt and R1441G cells (at least 10X more abundant than other isoforms). Now, di-18:1-BMP is roughly 4X more abundant in R1441G cells when compared to wt cells, while BMP is reduced by half in R1441G cells (light microscopy in Fig 1). Authors argue that light microscopy may only detects a so-called antibody accessible pool. What is this? And why would this pool decrease in R1441G cells when LAMP2 is higher? Alternatively, they argue that the anti-BMP antibody may be less specific and detect other analytes. As I had already mentioned, this makes no sense, since the observed signal is lower and not higher. If authors do not trust their light microscopy analysis, why show the data?

(2) Cells contain 3 LAMP2 isoforms. Which one is upregulated and/or secreted in exosomes?

(3) The new Fig S4A is far from convincing. How were cells fractionated and what are the gradients (not described in Methods)? CD63 (presumably endolysosomes) is spread over fractions 8 - 13. LRRK2 (fractions 8-9) does not copurify with CD63. The bulk of LRRK2 is at the bottom (presumably cytosol if this is a floatation gradient), and a minor fraction moves into the gradient. CLN5 is even less clear since the bulk is also at the bottom with a tiny fraction only between LRRK2 and CD63. Also, why do authors conclude that a considerable pool of newly synthesized CLN5 did not reach its final destination at the endolysosome and may instead be retained in the ER? Where is the ER on the gradient?

(4) Fig S4B shows blots of whole cell lysates from CTRL and LRRK2 mutant-derived fibroblasts: 6 lanes are shown but without captions, containing varying amounts of calnexin and CD63. In addition, the blots look very dirty. Where is CD63? Is it the minor band at ≈37 kD (as in Fig S4A)? Or the major band below the 50kD marker? What are the other bands on these blots? As a result, the quantification shown in the bar graph does not mean much.

(5) The cell content of 18.1-BMP is increased approx. 5X by BafA1 (Fig 6C) but amounts of 18.1-BMP secreted in EVs hardly changes (Fig 6E). Since BMP is mostly present as 18.1 isoform (22:6-BMP being only a minor species, Fig S1H), does it mean that BafA1 does not increase BMP secretion and/or only a minor fraction of total cellular BMP is secreted in exosomes?

Comments on revisions:

How come 0.2 mmol/L of 22:6 and 18:1 fatty acid both correspond to 65 µg/mL (Fig 4A)?

It is stated in the Legend of Fig4 that long (B-C) and short (D) chase time points are shown as fold change. There is no panel D in the figure.

---

## [Author Response]

The following is the authors’ response to the original reviews.

**eLife Assessment**
This useful study presents the potentially interesting concept that LRRK2 regulates cellular BMP levels and their release via extracellular vesicles, with GCase activity further modulating this process in mutant LRRK2-expressing cells. However, the evidence supporting the conclusions remains incomplete, and certain statistical analyses are inadequate. This work would be of interest to cell biologists working on Parkinson's disease.
**Reviewer #1 (Public review):**
Summary:Even though mutations in LRRK2 and GBA1 (which encodes the protein GCase) increase the risk of developing Parkinson's disease (PD), the specific mechanisms driving neurodegeneration remain unclear. Given their known roles in lysosomal function, the authors investigate how LRRK2 and GCase activity influence the exocytosis of the lysosomal lipid BMP via extracellular vesicles (EVs). They use fibroblasts carrying the PDassociated LRRK2-R1441G mutation and pharmacologically modulate LRRK2 and GCase activity.Strengths:The authors examine both proteins at endogenous levels, using MEFs instead of cancer cells. The study's scope is potentially interesting and could yield relevant insights into PD disease mechanisms.Weaknesses:Many of the authors' conclusions are overstated and not sufficiently supported by the data. Several statistical errors undermine their claims. Pharmacological treatment is very long, leading to potential off-target effects. Additionally, the authors should be more rigorous when using EV markers.

We thank the reviewer for these valuable observations. In the revised manuscript, we have addressed each of these points as follows:

(1) Conclusions and data support – We carefully revised our text throughout the manuscript to ensure that all conclusions are better supported by the presented data. For instance, we now explicitly state that while pharmacological modulation supports the regulatory role of LRRK2 activity in EV-mediated BMP release, we have softened our conclusions concerning the contribution of GCase in this model (see revised Results and Discussion sections).

(2) Statistical analyses – We reanalyzed experiments involving more than two groups and replaced simple t-tests with non-parametric Kruskal-Wallis tests followed by Dunn’s post hoc comparisons. This approach, described in the updated figure legends (e.g., Figure 2D-F and H-J), provides a more rigorous statistical framework that accounts for small sample sizes and variability typical of EV quantifications.

(3) Pharmacological treatment duration – Prolonged MLi-2 treatments have been extensively used in the field without evidence of significant off-target effects. Several studies, including Fell et al. (2015, J Pharmacol Exp Ther 355:397-409), De Wit et al. (2019, Mol Neurobiol 56:5273-5286), Ho et al. (2022, NPJ Parkinson’s Dis 8:115),Tengberg et al. (2024, Neurobiol Dis 202:106728), and Jaimon et al. (2025, Sci Signal 18:eads5761), have applied long-term (24-48 h) MLi-2 treatments at comparable concentrations without detecting toxicity or off-target alterations, including in MEFs (Ho et al., 2022; Dhekne et al., 2018, eLife 7:e40202). In our study, 48-hour incubations were necessary to sustain full LRRK2 inhibition throughout the extracellular vesicle (EV) collection period. EV biogenesis, BMP biosynthesis, and packaging into EVs are timedependent processes; therefore, extended incubation and collection periods (48 h) were required to allow downstream effects of LRRK2 inhibition on BMP production and release to manifest, and to obtain sufficient EV material for biochemical and lipidomic analyses. This experimental design also reflects our and others’ previous observations in humans and non-human primates, where urinary BMP changes are associated with chronic or subchronic LRRK2 inhibitor treatment (Baptista MAS, Merchant K, et al. Sci Transl Med. 2020, 12:eaav0820; Jennings D, et al. Sci Transl Med. 2022, 14:eabj2658; Maloney MT, et al. Mol Neurodegener. 2025, 20:89). Importantly, under these conditions, we did not observe significant changes in cell viability or morphology, supporting that the treatment was well tolerated. We have clarified this rationale in the revised Methods section to emphasize that the prolonged incubation reflects the experimental design for EV isolation rather than a requirement for achieving LRRK2 inhibition.

(4) EV markers – We and others have reported enrichment of Flotillin-1 and LAMP proteins in isolated small EV fractions (Kowal et al., 2016; Lu et al., 2018; Mathieu et al., 2021; Ferreira et al., 2022). Moreover, LAMP proteins have been reported to be more enriched in EVs of endolysosomal origin (Mathieu et al., 2021). To further strengthen this point, we performed new experiments using a CD63-pHluorin sensor combined with TIRF microscopy, which allowed real-time visualization of CD63-positive exosome release. These new data (now presented in Figure 7, Panels G-I; Videos 1 and 2) confirm increased CD63-positive EV release in LRRK2 mutant fibroblasts, which was reversed by LRRK2 inhibition with MLi-2. The CD63-positive compartment was also largely BMPpositive (new Figure 7D, F, G), reinforcing our conclusions and providing additional rigor in EV marker validation.

**Reviewer #2 (Public review):**
Summary:In this paper, the authors used MEFs expressing the R1441G mutant of leucine-rich repeat kinase 2 (LRRK2), a mutant associated with the early onset of Parkinson's disease. They report that in these cells LAMP2 fluorescence is higher but BMP fluorescence is lower, MVE size is reduced, and that MVEs contain less ILVs. They also report that LAMP2-positive EVs are increased in mutant cells in a process sensitive to LRRK2 kinase inhibition but are further increased by glucocerebrosidase (GCase) inhibition, and that total di-22:6-BMP and total di-18:1-BMP are increased in mutant LRRK2 MEFs compared to WT cells by mass spectrometry. They also report that LRRK2 kinase inhibition partially restores cellular BMP levels, and that GCase inhibition further increases BMP levels, and that in EVs from the LRRK2 mutant, LRRK2 inhibition decreases BMP while GCase inhibition has the opposite effect. Moreover, they report that the BMP increase is not due to increased BMP synthesis, although the authors observe that CLN5 is increased in LRRK2 mutant cells. Finally, they report that GW4869 decreases EV release and exosomal BMP, while bafilomycin A1 increases EV release. They conclude that LRRK2 regulates BMP levels (in cells) and release (via EVs). They also conclude that the process is modulated by GCase in LRRK2 mutant cells, and that these studies may contribute to the use of BMP-positive EVs as a biomarker for Parkinson's disease and associated treatments.Strengths:This is an interesting paper, which provides novel insights into the biogenesis of exosomes with exciting biomedical potential. However, I have comments that authors need to address to clarify some aspects of their study.Weaknesses:(1) The intensity of LAMP2 staining is increased significantly in cells expressing the R1441G mutant of LRRK2 when compared to WT cells (Figure 1C). Yet mutant cells contain significantly smaller MVEs with fewer ILVs, and the MVE surface area is reduced (Figure 1D-F). This is quite surprising since LAMP2 is a major component of the limiting membrane of late endosomes. Are other proteins of endo-lysosomes (eg, LAMP1, CD63, RAB7) or markers (lysotracker) also decreased (see also below)?

As referenced in our original manuscript, several previous studies have reported endolysosomal morphological and homeostatic defects in cells harboring pathogenic LRRK2 mutations. LAMP2 can be upregulated as part of a lysosomal biogenesis or stress response (e.g., via MiT/TFE transcription factors such as TFEB; Sardiello et al., Science 2009, 325:473-477), whereas ILV biogenesis is primarily controlled by ESCRT- and SMPD3-dependent pathways that are regulated independently of MiT/TFE-driven transcriptional programs. Indeed, Stuffers et al. (Traffic 2009, 10:925-937) demonstrated that depletion of key ESCRT subunits markedly inhibited ILV formation while concomitantly increasing LAMP2 expression, highlighting the mechanistic dissociation between LAMP2 abundance and ILV number. In our study, we observed a similar pattern in R1441G LRRK2 MEFs, in which elevated LAMP2 staining and protein levels occurred despite a reduction in MVE size and ILV number. We interpret this as a compensatory lysosomal biogenesis response.

Our revised manuscript now includes new immunofluorescence data for BMP, LAMP1 and CD63 (New Figure 7, Panels A-F) together with biochemical analysis of CD63 protein levels (New Supplemental Figure 4, Panel B) in human skin fibroblasts derived from healthy donors and LRRK2 G2019S PD patients. Quantitative analysis of these experiments revealed no statistically significant differences in total cellular levels of either LAMP1 or CD63 between groups. However, we observed a consistent decrease in BMP immunostaining intensity (New Figure 7, Panel A and B), in agreement with our findings in mouse fibroblasts. We therefore propose that the elevated LAMP2 expression observed in the engineered MEF clone expressing R1441G may reflect a cell type-specific effect, potentially linked to differential penetrance of LRRK2 signaling on the lysosomal biogenesis response. We have updated the Results and Discussion section of the manuscript to incorporate and clarify these findings.

(2) LRRK2 has been reported to interact with endolysosomal membranes. Does the R1441G mutant bind LAMP2- and/or BMP-positive membranes?

We agree that LRRK2 has been reported to associate dynamically with endolysosomal membranes, particularly under conditions of endolysosomal stress or damage (Eguchi T, et al. PNAS 2018, 115:E9115-E9124; Bonet-Ponce L, et al. Sci Adv. 2020, 6:eabb2454; Wang X, et al. Elife. 2023, 12:e87255).

Nevertheless, to explore whether LRRK2 associates with BMP-positive endolysosomes, we performed subcellular fractionation followed by biochemical analysis of endolysosomal fractions, since our available LRRK2 antibodies did not provide reliable immunofluorescence signals. These experiments were carried out using human skin fibroblasts derived from both healthy controls and Parkinson’s disease patients carrying the LRRK2-G2019S mutation. In both control and mutant fibroblasts, a pool of LRRK2 was detected in fractions positive for the BMP synthase CLN5 and the endolysosomal marker CD63 (New Supplementary Figure 4, Panel A), supporting the localization of LRRK2 to endolysosomal membranes that are likely BMP-enriched. Our manuscript’s Results and Methods sections have been updated accordingly.

Does the mutant affect endolysosomes?

As referenced in our original manuscript, several studies have reported that pathogenic LRRK2 mutations can lead to endolysosomal defects. Consistent with these reports, we also observed morphological alterations in endolysosomes of cells expressing mutant LRRK2, including reduced MVE size and fewer ILVs, as shown in Figure 1D–F. These observations are in agreement with previously described phenotypes associated with pathogenic LRRK2 variants. Furthermore, in mutant LRRK2 MEFs, and now in humanderived fibroblasts (see new Figure 7, Panel A and B), we observed a decrease in BMP immunostaining signal.

(3) Immunofluorescence data indicate that BMP is decreased in mutant LRRK2expressing cells compared to WT (Figure 1A-B), but mass spec data indicate that di-22:6BMP and di-18:1-BMP are increased (Figure 3). Authors conclude that the BMP pool detected by mass spec in mutant cells is less antibody-accessible than that present in wt cells, or that the anti-BMP antibody is less specific and that it detects other analytes. This is an awkward conclusion, since the IF signal with the antibody is lower (not higher): why would the antibody be less specific? Could it be that the antibody does not see all BMP isoforms equally well? Moreover, the observations that mutant cells contain smaller MVEs (Figure 1D-F) with fewer ILVs are consistent with the IF data and reduced BMP amounts. This needs to be clarified.

As previously reported by us (Lu et al., J Cell Biol 2022;221:e202105060) and others (Berg AL, et al. Cancer Lett. 2023, 557:216090), discrepancies can occur between BMP levels detected by immunofluorescence and those quantified by mass spectrometry. This is because immunostaining reflects the pool of antibody-accessible BMP, whereas lipidomics measures the total cellular content of all BMP molecular species, irrespective of their distribution or accessibility.

We agree that the anti-BMP antibody may not detect all BMP isoforms equally well. Differences in acyl chain composition (such as the degree of saturation or chain length) can alter the stereochemistry of BMP and, consequently, epitope accessibility to antibody binding.

In addition, in a personal communication with Monther Abu-Remaileh (Stanford University), we were informed that the antibody may also cross-react with other lipid species in endolysosomes. Nevertheless, since there is no formal evidence supporting this, we have removed the sentence in the Discussion section stating “Alternatively, the antibody may also detect non-BMP analytes” to avoid any potential misinterpretations. In its place, we have added a short statement noting that “not all BMP isoforms may be detected equally well”.

Mass spectrometry data are only shown for two BMP species (di-22:6, di-18:1). What are the major BMP isoforms in WT cells? The authors should show the complete analysis for all BMP species if they wish to draw quantitative conclusions about the amounts of BMP in wt and mutant cells. Finally, BMP and PG are isobaric lipids. Fragmentation of BMPs or PGs results in characteristic fingerprints, but the presence of each daughter ion is not absolutely specific for either lipid. This should be clarified, e.g., were BMP and PG separated before mass spec analysis? Was PG affected? The authors should also compare the BMP data with mass spec data obtained with a control lipid, e.g., PC.

Regarding BMP isoforms, our targeted UPLC-MS/MS analyses revealed that 2,2′-di-22:6-BMP (sn2/sn2′) and 2,2′-di-18:1-BMP (sn2/sn2′) are the predominant BMP isoforms in MEF cells, consistent with previous reports showing docosahexaenoyl (22:6; DHA) and oleoyl (18:1) BMP as the most abundant isoforms. Across diverse mammalian cells and tissues, BMP typically exhibits a fatty acid composition dominated by oleoyl, with polyunsaturated fatty acids (particularly DHA) also contributing substantially. Enrichment of DHA-containing BMP species has been observed in multiple systems, including rat uterine stromal cells, PC12 cells, THP-1 and RAW macrophages, as well as in rat and human liver. This consistent presence of oleoyl- and docosahexaenoyl-containing BMP species across tissues indicates that these acyl chains are conserved features influencing the lipid’s structural and functional characteristics (Kobayashi et al. J Biol Chem, 2002; Hullin-Matsuda et al. Prostaglandins Leukotriens Essent Fatty Acids, 2009; Thompson et al. Int J Toxicol. 2012; Delton-Vandenbroucke et al. J Lipid Res, 2019).

Nevertheless, we have included a Table (Panel H in updated Supplemental Figure 1) showing other BMP species that were also detected in our lipidomics analysis. Overall, dioleoyl (18:1)- and di-docosahexaenoyl (22:6)-BMP species were the most abundant in MEF cells, whereas di-arachidonoyl (20:4)- and di-linoleoyl (18:2)-BMP isoforms were present at lower levels. Consistently, R1441G LRRK2 MEFs displayed higher levels of dioleoyl- and di-docosahexaenoyl-BMP compared with WT cells, and these elevations were reduced following LRRK2 kinase inhibition with MLi-2. Data from three independent representative experiments are shown, and the manuscript has been revised accordingly to include these results.

Regarding the separation of BMP and PG species, we confirm that BMP and PG were chromatographically resolved prior to MS/MS detection using a validated UPLC-MS/MS method developed by Nextcea, Inc PG exhibits a substantially longer LC retention time than BMP, ensuring complete baseline separation. This approach (established by Nextcea nearly two decades ago and later validated through a multi-year collaboration with the U.S. FDA to clinically qualify di-22:6-BMP as a biomarker) prevents any ambiguity arising from the isobaric nature of BMP and PG species. No changes in PG levels were detected under any experimental conditions.

Finally, we employed isotope-labeled BMP as an internal standard to ensure robust normalization across samples. These additional details and references cited above have been included in the revised Methods and References sections to further clarify the analytical rigor of our lipidomics workflow.

(4) It is quite surprising that the amounts of labeled BMP continue to increase for up to 24h after a short 25min pulse with heavy BMP precursors (Figure 4B).

In these isotope-labeling experiments, it is important to note (as described in our original manuscript) that two distinct pools of metabolically labeled BMP species were detected: semi-labeled BMP (with only one heavy isotope-labeled fatty acyl chain) and fully-labeled BMP (with both fatty acyl chains labeled). We consider the fully-labeled BMP pool to provide the most reliable readout for BMP turnover, as it showed a rapid decline after a 1h chase (decreasing by more than 50% within 8 h in all conditions), reaching its lowest levels at the end of the 48-h chase period.

The apparent increase in semi-labeled BMP species over time may be explained by continued incorporation of labeled precursors following the initial pulse. Specifically, once existing semi-labeled and fully-labeled BMP molecules are degraded by PLA2G15 (Nyame K, et al. Nature 2025, 642:474-483), the resulting isotope-labeled lysophosphatidylglycerol (LPG) and fatty acids could be recycled and re-enter a new round of BMP biosynthesis, leading to a gradual accumulation of semi-labeled BMP such as di-18:1-BMP. Why would this reasoning not also apply to the fully-labeled species? Once the pulse is completed, newly incorporated non-labeled fatty acyl chains present in the cellular pool can compete with labeled ones during subsequent rounds of lipid remodeling or synthesis. As a result, the probability of generating semi-labeled BMP molecules becomes higher than that of forming fully-labeled species. Consistent with this, our data show an increase in only semi-labeled BMP species (but not in fully-labeled ones) up to 24 hours after the pulse. We have added a clarification regarding this point in the revised manuscript.

(5) It is argued that upregulation of CLN5 may be due to an overall upregulation of lysosomal enzymes, as LAMP2 levels were also increased (Figure 2A, C, E). Again, this is not consistent with the observed decrease in MVE size and number (Figure 1D-F). As mentioned above, other independent markers of endo-lysosomes should be analyzed (eg, LAMP1, CD63, RAB7), and/or other lysosomal enzymes (e.g. cathepsin. D).

Our revised manuscript now includes new immunofluorescence data for BMP, LAMP1 and CD63 (New Figure 7, Panels A-F) together with biochemical analysis of CD63 protein levels (New Supplemental Figure 4, Panel B) in human skin fibroblasts derived from healthy controls and LRRK2 G2019S PD patients. Quantitative analysis of these experiments revealed no statistically significant differences in total cellular levels of either LAMP1 or CD63 between groups. However, our results consistently show increased CLN5 protein levels in both mouse and human fibroblast cell lines harboring pathogenic LRRK2 mutations. Upregulation of CLN5 may reflect a compensatory effect from loss of BMP via EV exocytosis. As discussed above, the elevated LAMP2 signal observed in the engineered MEF clone expressing R1441G could represent a cell type-specific effect, potentially linked to differential penetrance of LRRK2 signaling on the lysosomal biogenesis response. Our Results and Discussion sections have been updated accordingly.

(6) The authors report that the increase in BMP is not due to an increase in BMP synthesis (Figure 4), although they observe a significant increase in CLN5 (Figure 5A) in LRRK2 mutant cells. Some clarification is needed.

In our original manuscript, we proposed that although CLN5 protein levels are increased in R1441G LRRK2 MEFs, the absence of significant changes in BMP synthesis rates (Figure 4B, C) may reflect either limited substrate availability or that CLN5 is already operating near its maximal enzymatic capacity. Our new subcellular fractionation data (new Figure 7, Panel A) further indicate that, despite a relative increase in total CLN5 levels in G2019S LRRK2 human fibroblasts, the amount of CLN5 associated with endolysosomes remains comparable between mutant LRRK2 and control cells. This suggests that a considerable fraction of upregulated CLN5 may not localize to endolysosomes, potentially accumulating in the endoplasmic reticulum due to enhanced translation or impaired trafficking. Unfortunately, the available anti-CLN5 antibody did not yield reliable immunofluorescence signals, preventing us from directly confirming this possibility. Nevertheless, in light of our new data (new Supplemental Figure 4A), we have included a clarification in the revised manuscript discussing this possibility as well.

(7) Authors observe that both LAMP2 and BMP are decreased in EVs by GW4869 and increased by bafilomycin (Figure 6). Given my comments above on Figure 1, it would also be nice to illustrate/quantify the effects of these compounds on cells by immunofluorescence.

We appreciate the reviewer’s suggestion. We have previously published immunofluorescence data showing increased BMP accumulation in endolysosomes following treatment with bafilomycin A1 Lu A, et al. J Cell Biol. 2009, 184:863-879. However, in the present study, our lipidomics analyses revealed a decrease in both di22:6-BMP and di-18:1-BMP species in cells treated with this compound. As discussed above, this apparent discrepancy likely reflects methodological differences between immunofluorescence, which detects only antibody-accessible BMP pools, and lipidomics, which quantifies total cellular BMP content.

Moreover, in a recent study (Andreu Z, et al. Nanotheranostics 2023, 7:1-21), BMP levels were analyzed by immunofluorescence in cells treated with spiroepoxide, a potent and selective irreversible inhibitor of nSMase (different from GW4869) known to block EV release. Spiroepoxide-treated cells showed decreased BMP immunostaining; a result that, again, does not align with mass spectrometry data revealing increased cellular BMP levels upon GW4869 treatment. Notably, in that study, spiroepoxide was used instead of GW4869 because the intrinsic autofluorescence of GW4869 could potentially interfere with the immunofluorescence BMP signal.

We therefore consider lipidomics measurements to provide a more reliable and quantitative representation of BMP dynamics under these conditions.

**Reviewer #1 (Recommendations for the authors):**
Major concerns:(1) 48 h for MLi2 treatment seems too long. LRRK2 kinase activity is inhibited with much shorter incubation times. The longer the incubation, the more likely off-target effects are. The authors should repeat these experiments with 1-2 h of MLi2.

We thank the reviewer for this valuable comment. We acknowledge that MLi-2 is a potent and selective LRRK2 kinase inhibitor that achieves near-complete target engagement within a few hours of treatment. However, prolonged exposure has been widely used in the field without evidence of significant off-target effects. Several studies, including Fell et al. (2015, J Pharmacol Exp Ther 355:397-409), De Wit et al. (2019, Mol Neurobiol 56:5273-5286), Ho et al. (2022, NPJ Parkinson’s Dis 8:115), Tengberg et al. (2024, Neurobiol Dis 202:106728), and Jaimon et al. (2025, Sci Signal 18:eads5761), have employed long-term (24-48 h) MLi-2 treatments at comparable concentrations without detecting toxicity or off-target alterations, including in MEFs (Ho et al., 2022; Dhekne et al., 2018, eLife 7:e40202).

In our study, 48-hour incubations were necessary to sustain full LRRK2 inhibition throughout the extracellular vesicle (EV) collection period. EV biogenesis, BMP biosynthesis, and packaging into EVs are time-dependent processes; therefore, extended incubation and collection periods (48 h) were required to allow downstream effects of LRRK2 inhibition on BMP production and release to manifest, and to obtain sufficient EV material for biochemical and lipidomic analyses. This experimental design also reflects our and others’ previous observations in humans and non-human primates, where urinary BMP changes are associated with chronic or subchronic LRRK2 inhibitor treatment (Baptista MAS, Merchant K, et al. Sci Transl Med. 2020, 12:eaav0820; Jennings D, et al. Sci Transl Med. 2022, 14:eabj2658; Maloney MT, et al. Mol Neurodegener. 2025, 20:89). Importantly, under these conditions, we did not observe significant changes in cell viability or morphology, supporting that the treatment was well tolerated.

We have clarified this rationale in the revised Methods section to emphasize that the prolonged incubation reflects the experimental design for EV isolation rather than a requirement for achieving LRRK2 inhibition.

(2) Is there a reason why the authors don't include CD81, CD63, and Syntenin-1 in their study as an EV marker? Using solely Flotilin-1 does not seem to be enough to justify their claims.

We actually used not only Flotillin-1 but also LAMP2 as EV markers in our study. While both Flotillin-1 and LAMP2 detection on EVs may vary depending on the cell type, we and others have reported enrichment of Flotillin-1 and LAMP proteins in isolated small EV fractions (Kowal et al., 2016; Lu et al., 2018; Mathieu et al., 2021; Ferreira et al., 2022). In particular, one of these studies reported that “LAMP1-positive subpopulations of EVs represent MVB/lysosome-derived exosomes, which also contain syntenin-1.” Therefore, our choice of EV markers (LAMP2 and Flotillin-1) is consistent with those previously and reliably used to characterize small EVs.

Nevertheless, to further address the reviewer’s concern, we performed additional experiments using a CD63-based fluorescence sensor (CD63-pHluorin), which, combined with TIRF microscopy, enables real-time visualization of CD63-positive exosome release. These experiments were conducted in control and LRRK2-mutant fibroblasts, and the data are presented in new Figure 7 (Panels G-I; Videos 1 and 2). We have also included all relevant references and clarified this point in the revised manuscript.

(3) Indeed, to quantify the amount of certain proteins in EVs, the authors should normalize them by CD63 or CD81.

Protein normalization in isolated EV fractions is indeed challenging. Although tetraspanins such as CD63 and CD81 are commonly enriched in EVs, their abundance can vary considerably across EV subpopulations, cell types, and experimental conditions, making them unreliable as universal normalization markers (Théry et al., J Extracell Vesicles, 2018; Margolis & Sadovsky, Nat Rev Mol Cell Biol, 2019). Current guidelines from the International Society for Extracellular Vesicles (ISEV), as described in the Minimal Information for Studies of Extracellular Vesicles 2018 (MISEV2018; Théry C, et al. JExtracell Vesicles. 2018, 7:1535750) and updated in MISEV2024 (Welsh JA, et al. J Extracell Vesicles. 2024, 13:e12404), recommend reporting multiple EV markers rather than relying on a single protein for normalization. They also suggest ensuring comparable experimental conditions by using the same number of cells at the start of the experiment and normalizing EV data to cell number or whole-cell lysate protein content at the end of the experiment, among other approaches.

In our study, we normalized EV data to whole-cell lysate (WCL) protein content, as this approach accounts for differences in EV production due to variations in cell number or treatment conditions and is commonly used in the field (Kowal et al., PNAS, 2016; Mathieu et al., Nat Commun, 2021). We also included Flotillin-1 and LAMP2 as EV markers, both of which have been validated as molecular markers of small EV subpopulations.

(4) Hyper normalization in WB quantification in Figure 2E-G is statistically incorrect, as it assumes that one group (in this case, R1441G ctrl) has no variability at all, which is not biologically possible. The authors should repeat the quantification without hypernormalizing one of their groups. This issue is prevalent across the whole manuscript.

We understand the concern regarding “hyper-normalization” (i.e., expressing all values relative to one condition set to 1), which may mask variability in the reference group. However, it is standard practice in immunoblotting analysis to express data relative to a control condition for comparison, as variations in membrane transfer, exposure time, and signal development can differ across blots. In our case, the data are expressed as relative levels (arbitrary units) rather than absolute quantitative values. To facilitate comparison between datasets and account for inter-experimental variation, we continued to express values relative to the mutant LRRK2 MEF condition.

On the other hand, in lipidomics experiments, despite using the same number of seeded cells and identical extraction and analysis protocols, minor biological and technical variability was observed across independent replicates. This variability is inherent to the experimental system and is now explicitly represented in the new table included in Supplemental Figure 1F, which compiles three independent representative lipidomics experiments showing quantitative BMP levels across different conditions.

(5) The authors perform a t-test in Figure 2E-G when comparing more than 2 groups, which is wrong. The authors should use a two-way ANOVA as they are comparing genotype and treatment.

We appreciate the reviewer’s comment and agree with this observation. The MLi-2 and CBE experiments were performed independently and in separate experimental runs; therefore, we have reanalyzed these datasets separately rather than combining them in a two-way ANOVA. To properly compare more than two groups within each dataset, we have now applied a Kruskal-Wallis test followed by an uncorrected Dunn’s post hoc test (Figure 2 D-F and H-J). This non-parametric approach is more appropriate for our data structure, as EV experiments are usually subject to high variability and immunoblot quantifications involving small sample sizes (n≈6) do not always meet the assumptions of normality or equal variance. The Kruskal-Wallis test does not assume normality or equal variances, making it more robust for small, variable biological datasets. The statistical analyses and figure legend have been updated in the revised manuscript accordingly.

In addition, since our CBE treatments yielded statistically non-significant data, we have softened our conclusions throughout the manuscript concerning the contribution of GCase activity to EV-mediated BMP release modulation.

(6) There is a very strong reduction in flotillin-1 in R1441G cells vs WT (Figure 2G) in the EV fraction. That reduction is further exacerbated with MLi2, which likely means it is not kinase activity dependent. Can the authors comment on that?

We agree with the reviewer that Flotillin-1 showed a different behavior compared with LAMP2 in these experiments. As recommended by the MISEV guidelines (Théry C, et al. J Extracell Vesicles. 2018; 7:1535750; Welsh JA, et al. J Extracell Vesicles. 2024, 13:e12404), it is important to analyze more than one EV-associated protein marker. We examined LAMP2, which, together with LAMP1, has been reported to be specifically enriched in EVs of endolysosomal origin (exosomes; Mathieu et al., Nat Commun. 2021, 12:4389). In contrast, Flotillin-1 is also associated with small EVs but may represent a distinct EV subpopulation from those positive for LAMP proteins (Kowal J, et al. PNAS 2016, 113:E968-E977).

Nevertheless, the biochemical analysis of isolated EV fractions was complemented by our lipidomics data and, in the revised version, by TIRF microscopy analysis of exosome release in control and G2019S LRRK2 human fibroblasts (new Figure 7, Panels G-I; Videos 1 and 2). In this analysis, we confirmed increased exocytosis of CD63-pHluorin– positive endolysosomes in G2019S LRRK2 human fibroblasts compared to controls, an effect that was reversed by MLi-2 treatment. The CD63-pHluorin–positive compartment of these cells was also largely positive for BMP (new Figure 7G). Collectively, these findings further support the regulatory role of LRRK2 activity in EV-mediated BMP secretion.

(7) In Figure 2C, the authors should express that the LAMP2-EV and flotillin-1 EV fractions from the WB are highly exposed. As presently presented, it is slightly misleading.

We thank the reviewer for this comment. In EV preparations, the amount of protein recovered is typically very low. Therefore, although we loaded all the EV protein obtained from each sample, the immunoblots for LAMP2 and Flotillin-1 in EV fractions required longer exposure times to visualize clear signals across all conditions. We have now indicated in the corresponding figure legend that these EV blots are long-exposure blots to facilitate signal detection and avoid any potential misunderstanding.

(8) If Figure 2C and D are from two different experiments, they should not be plotted together in Figure 2E-G. You cannot compare the effect of MLi2 vs CBE if done in completely different experiments.

We appreciate the reviewer’s comment and agree with this observation. The MLi-2 and CBE experiments were performed independently and in separate experimental runs; therefore, we have reanalyzed these datasets separately rather than combining them in a two-way ANOVA. To properly compare more than two groups within each dataset, we have now applied a Kruskal-Wallis test followed by an uncorrected Dunn’s post hoc test (Figure 2 D-F and H-J). This non-parametric approach is more appropriate for our data structure, as EV experiments are usually subject to high variability and immunoblot quantifications involving small sample sizes (n≈6) do not always meet the assumptions of normality or equal variance. The Kruskal-Wallis test does not assume normality or equal variances, making it more robust for small, variable biological datasets. The revised statistical analyses and figure legends have been updated accordingly in the manuscript.

(9) The authors state that "For the R1441G MEF cells, MLi-2 decreased EV concentration while CBE increased EV particles per ml, in agreement with the effects observed in our biochemical analysis." As Figure S1D shows no statistical significance, the authors don't have sufficient evidence to make this claim.

We apologize for this overstatement. We have revised the text to clarify that, although the differences did not reach statistical significance, a consistent trend toward decreased EV concentration upon MLi-2 treatment and increased EV release following CBE treatment was observed in R1441G MEF cells.

(10) "Altogether, given that BMP is specifically enriched in ILVs (which become exosomes upon release), the data presented above support our biochemical analysis (Figure 2C, D, F) and suggest a role for LRRK2 and GCase in modulating BMP release in association with LAMP2-positive exosomes from MEF cells." As Figure 3E shows no statistical difference of BMP on EVs upon CBE treatment, this sentence is not accurate and should be reframed. Furthermore, the authors claim an increase in EV-LAMP2 in R1441G cells compared to WT, however, the amount of BMP in EVs of R1441G cells vs WT is unchanged with a non-significant reduction. This contradiction does not support the authors' conclusions and really puts into question their whole model.

We thank the reviewer for this observation. After reanalyzing our biochemical data from isolated EV fractions (see new Panels D-F and H-J) using an improved statistical approach, we found that although EV-associated LAMP2 levels were consistently elevated in untreated R1441G LRRK2 MEFs compared to WT cells, CBE treatment only produced a non-significant trend toward increased EV-associated LAMP2 compared to untreated R1441G LRRK2 cells. Accordingly, we have revised the sentence to read as follows:

“Altogether, given that BMP is specifically enriched in ILVs (which become exosomes upon release), the data presented above support our biochemical analysis (Figure 2C, E, G, I) and suggest that LRRK2 activity regulates BMP release in association with LAMP2positive exosomes, whereas GCase activity appears to have a more variable effect under the tested conditions.”

We also agree with the reviewer that, in our MEF model, the amount of BMP in EVs of R1441G cells vs WT is unchanged with a non-significant reduction. However, pharmacological modulation supports our conclusion that BMP release is modulated by LRRK2 activity. Specifically, treatment with the LRRK2 inhibitor MLi-2 decreased EVassociated BMP and LAMP2 levels in R1441G LRRK2 MEFs, and our new data (new Figure 7, Panel G-I; Videos 1 and 2) show increased exocytosis of CD63-pHluorin– positive endolysosomes in G2019S LRRK2 human fibroblasts compared to controls, an effect that was reversed by MLi-2 treatment. The CD63-pHluorin–positive compartment of these cells was also largely positive for BMP (new Figure 7G).

In light of the reviewer’s comment about CBE treatment, we have softened our conclusions throughout the manuscript concerning the contribution of GCase activity in this model.

(11) In Figure 5, 16 h of MLi2 treatment is too long and can lead to off-target effects. I would advise reducing it to 1-4 h.

Prolonged MLi-2 treatments have been extensively used in the field without evidence of significant off-target effects. Several studies, including Fell et al. (2015, J Pharmacol Exp Ther 355:397-409), De Wit et al. (2019, Mol Neurobiol 56:5273-5286), Ho et al. (2022, NPJ Parkinson’s Dis 8:115), Tengberg et al. (2024, Neurobiol Dis 202:106728), and Jaimon et al. (2025, Sci Signal 18:eads5761), have applied long-term (24-48 h) MLi-2 treatments at comparable concentrations without detecting toxicity or off-target alterations, including in MEFs (Ho et al., 2022; Dhekne et al., 2018, eLife 7:e40202). Moreover, the data presented in Figure 5 demonstrate a reduction in CLN5 protein levels in both MEFs and human fibroblasts following MLi-2 treatment, confirming the specificity of the observed effects in LRRK2 mutant cells.

(12) "Our data suggest that BMP is exocytosed in association with EVs and that LRRK2 and GCase activities modulate BMP secretion." Again, cells carrying the R1441G mutation have the same amount of BMP in EVs than WT. This sentence is not factually accurate. Accordingly, CBE did not change the amount of BMP in EVs.

We thank the reviewer for this observation and agree that, in our MEF model, the amount of BMP in EVs from R1441G LRRK2 cells is comparable to that observed in WT cells. However, pharmacological modulation supports our conclusion that BMP release is modulated by LRRK2 activity. Specifically, treatment with the LRRK2 inhibitor MLi-2 decreased EV-associated BMP levels in R1441G LRRK2 MEFs, and our new data (new Figure 7G-I; Videos 1 and 2) show increased exocytosis of CD63-pHluorin–positive endolysosomes in G2019S LRRK2 human fibroblasts compared to controls, an effect that was reversed by MLi-2 treatment. The CD63-pHluorin–positive compartment of these cells was also largely positive for BMP (new Figure 7G). These findings further support the regulatory role of LRRK2 activity in EV-mediated BMP secretion. In addition, in light of the reviewer’s comment about CBE treatment, we have softened our conclusions throughout the paper concerning the contribution of GCase activity in this model.

(13) Figure 6; EV release should have been monitored by more accurate markers such as CD63 and CD81.

We thank the reviewer for this comment. We and others (Kowal et al., 2016; Lu et al., 2018; Mathieu et al., 2021; Ferreira et al., 2022) have reported enrichment of Flotillin-1 and LAMP proteins in isolated small EV fractions. In particular, one of these studies (Mathieu et al., Nat Commun. 2021), in which bafilomycin A1 was also used (to boost exosome release), reported that “LAMP1-positive subpopulations of EVs represent MVB/lysosome-derived exosomes, which also contain syntenin-1.” Altogether, our choice of EV markers (LAMP2 and Flotillin-1) is consistent with those previously and accurately used to characterize EVs. We have now included all relevant references in the revised manuscript to further clarify this point.

(14) Figure 6 suggests that exosomal BMP is controlled by EV release. I would think that is rather obvious.

We agree that the finding that exosomal BMP release is influenced by EV secretion may appear “obvious.” However, our intention in Figure 6 was to provide direct experimental evidence confirming this relationship using pharmacological modulators of EV release. Specifically, inhibition of EV secretion with GW4869 reduced exosomal BMP levels, whereas stimulation with bafilomycin A1 increased them. These data were important to establish a causal link between EV trafficking and BMP export, thereby validating our model and supporting the interpretation that LRRK2 regulates BMP homeostasis through EV-mediated exocytosis, which is further modulated, to some extent, by GCase activity.

Minor concerns:(1) Figure 1: Change colors to be color blind friendly.

We thank the reviewer for this helpful suggestion. We have adjusted the colors in Figure 1 to be color-blind friendly. In addition, we have applied the same color-blind friendly palette to the new immunofluorescence data presented in new Figure 7, Panel A and D.

(2) More consistency on "Xmin" vs "X min" would be appreciated.

We thank the reviewer for this observation. We have revised the manuscript to ensure consistent formatting of time indications throughout the text and figures, using the standardized format “X min.”

**Reviewer #2 (Recommendations for the authors):**
(1) Figure 2C-D. Were equal amounts of protein loaded in each lane?

Equal protein amounts were loaded in lanes corresponding to whole-cell lysate (WCL) fractions and normalized based on α-Tubulin levels.

For the extracellular vesicle (EV) fractions, all protein recovered from EV pellets after isolation was loaded. In all EV-related experiments, we seeded the same number of EVproducing cells per condition, and the resulting EV-derived data (from both immunoblotting and lipidomics analyses) were normalized to the corresponding whole cell lysate (WCL) protein content to ensure comparability across conditions.

All these technical details have been included in the Materials section of our revised manuscript.

(2) The authors refer to the papers of Medoh et al (ref 43) and Singh et al. (44) for the key role of CLN5 in the BMP biosynthetic pathway. However, Medoh et al reported that CLN5 is the lysosomal BMP synthase. In contrast, Singh et al. reported that PLD3 and PLD4 mediate the synthesis of SS-BMP, and did not find any role for CLN5.

To avoid any confusion or misinterpretation of our findings regarding CLN5 and given that we do not analyze PLD3 or PLD4 in our study, we have decided to replace the reference to Singh et al. with Bulfon D. et al. (Nat. Commun. 2024, 15:9937) instead. This last work, conducted by an independent group distinct from the one that originally described CLN5, also validated CLN5 as the sole BMP synthase in cells.

Also, authors mention that bafilomycin A1 (B-A1) dramatically boosts EV exocytosis, referring to Kowal et al., 2016 (ref 35) and Lu et al., 2018 (ref 45). However, this is not shown in Kowal et al.

We thank the reviewer for pointing out this mistake. We apologize for the incorrect citation and have now corrected the reference. The statement regarding the effect of bafilomycin A1 on EV exocytosis now appropriately refers to Mathieu et al., 2021 and Lu et al., 2018.

(3) Page 7, it is stated that "No statistically significant differences in intracellular BMP levels were observed in WT LRRK2 MEFs upon LRRK2 or GCase inhibition(Supplemental Figure 1D, E)". The authors probably mean "Supplemental Figure 1F, G"

We thank the reviewer for noting this error. We have corrected the text to refer to panels F and G of Supplemental Figure 1, which correspond to the relevant data. We have also revised the reference to panel I of Supplemental Figure 1 accordingly.